# DISTILLED PROTEIN BACKBONE GENERATION

## ABSTRACT

Diffusion- and flow-based generative models have recently demonstrated strong performance in protein backbone generation tasks, offering unprecedented capabilities for *de novo* protein design. However, while achieving notable performance in generation quality, these models are limited by their generating speed, often requiring hundreds of iterative steps in the reverse-diffusion process. This computational bottleneck limits their practical utility in large-scale protein discovery, where thousands to millions of candidate structures are needed. To address this challenge, we explore the techniques of score distillation, which has shown great success in reducing the number of sampling steps in the vision domain while maintaining high generation quality. However, a straightforward adaptation of these methods results in unacceptably low designability. Through extensive study, we have identified how to appropriately adapt Score identity Distillation (SiD), a state-of-the-art score distillation strategy, to train few-step protein backbone generators which significantly reduce sampling time, while maintaining comparable performance to their pretrained teacher model. In particular, multistep generation combined with inference time noise modulation is key to the success. We demonstrate that our distilled few-step generators achieve more than a 20-fold improvement in sampling speed, while achieving similar levels of designability, diversity, and novelty as the Proteína teacher model. This reduction in inference cost enables large-scale *in silico* protein design, thereby bringing diffusion-based models closer to real-world protein engineering applications.

## 1 INTRODUCTION

The field of *de novo* protein design revolves around the goal of generating new proteins that are unobserved in nature, with functions designed for specific biological objectives (Huang et al., 2016). Traditional protein design methods work by modifying existing proteins using techniques like directed evolution (Dougherty & Arnold, 2009), which requires significant domain expertise and extensive laboratory experimentation. With the rapid development of deep generative models and their proven success in tasks such as image generation, it is natural for researchers to attempt to apply the same generative models to the protein generation field.

Applying an image generation method to proteins is not straightforward due to several reasons. First, proteins do not have a fixed canonical orientation. Adaptations to the neural network have to be made to account for the SE(3)-equivariance. Second, a model must be able to respect the physical constraints and track the pairwise distances between residues. Third, errors in the local or global structure can lead to completely undesignable structures (Eguchi et al., 2022; Anand & Achim, 2022), whereas comparable levels of error for image models might only result in unrealistic images. The advancements in AlphaFold2 (Jumper et al., 2021) such as the invariant point attention (IPA) and the triangle multiplicative and self-attention layers have facilitated SE(3)-equivariant reasoning on residue frames and modeling the pairwise distances for residues in a protein structure. These architectural innovations led to earlier works such as RFDiffusion (Watson et al., 2023), Chroma (Ingraham et al., 2023), FrameDiff (Yim et al., 2023b), and Genie (Lin & AlQuraishi, 2023), all of which have demonstrated that diffusion models can be successfully applied to generate high-quality, designable protein backbone structures. Importantly, backbones generated by these models are not mere interpolations or reproductions of natural proteins but can encode genuinely novel folds, opening up a new frontier in *de novo* protein design.

However, a critical obstacle remains: **sampling speed**. The above-mentioned diffusion-based models often require hundreds or even 1000 iterative sampling steps. The problem is worsened by the computationally expensive IPA and triangle layers common in these protein structure generation models. This inefficiency in *de novo* structure generation is a major barrier to large-scale protein design: biologists often need to explore vast protein structure spaces, generating tens of thousands of candidate backbones to evaluate stability, binding affinity, or other functional properties. Slow generation becomes a bottleneck in this iterative generate–test exploration cycle, limiting the throughput and delaying the discovery of promising structures. Thus, accelerating protein structure generation is not only a matter of efficiency but a prerequisite for practical usage.

Some efforts to speed up the sampling process have been made recently. For example, flow-based FrameFlow (Yim et al., 2023a) showed significant improvements in sampling speed compared to its diffusion-based predecessor FrameDiff (Yim et al., 2023b). Proteína (Geffner et al., 2025) utilized flow matching with an SDE solver and showed that even without an equivariant architecture and IPA and triangle layers, their model can still achieve state-of-the-art performance in terms of designability and sampling speed. Nonetheless, Proteína requires 400 sampling steps, which leaves significant room for improvement if the right acceleration technique can be applied.

In the image domain, multiple methods have been developed to reduce the required number of sampling steps and speed up generation (Ma et al., 2024; Shen et al., 2025). In particular, distillation methods have shown remarkable performance by distilling the knowledge of a pretrained score estimation network and train a one- or few-step generator (Fan et al., 2025). Recent efforts like the score identity distillation (SiD) method (Zhou et al., 2024) and its adversarial enhancement in Zhou et al. (2025b) have shown state-of-the-art image generation quality with a distilled one-step generator, while the recent work in Zhou et al. (2025a) have extended the idea to few-step generators to further improve the results.

Another major challenge in protein structure generation is the model's sensitivity to local structural errors. Designable proteins require precise backbone geometries. Small inaccuracies can render a generated structure structurally invalid. To mitigate this, most protein generation methods apply noise rescaling at inference time, effectively lowering the sampling temperature to trade diversity for designability. While effective for standard diffusion sampling, this practice introduces a critical mismatch: distillation methods rely heavily on the original noise schedule learned during training, and rescaling breaks the alignment between the forward and reverse processes. Therefore, applying an off-the-shelf distillation method straightaway would fail catastrophically and one-step generators suffer from generation quality.

In this work, we develop a distillation framework that can be applied to both diffusion- and flow-based models, based on SiD (Zhou et al., 2024). We show that a one-step generator distilled under this framework does perform poorly and almost never produces a designable structure, due to the lack of noise rescaling. After extensive efforts to improve its performance have failed, we identified the necessary steps toward distilling protein backbone generation models: training few-step generators and sampling with low temperature. We investigate the impact of noise rescaling at inference and demonstrate that, with the appropriate noise scaling factor, our distilled 16-step generator can beat the pretrained teacher model in terms of the designability, novelty, and other metrics, while achieving a more than 20-fold reduction in sampling time. If higher designability is preferred, we can increase the number of steps even further. To the best of our knowledge, this is the first demonstration of distilling a protein backbone generator to achieve such speedups while also improving performance. The distilled model is particularly helpful in the context of *in silico* protein design, where the ability to fast generate large libraries of candidates is crucial for downstream evaluation and experimental validation. By reducing the sampling time from hundreds of steps to as few as 16, our approach makes it feasible to scale protein generation to the thousands to millions of structures often needed for practical discovery workflows. This enables tighter integration of generative models with iterative generate–test exploration cycles, and enhances the efficiency of large-scale screening.

## 2 BACKGROUND AND RELATED WORK

**Protein structure representation.** Proteins are often represented as the combination of sequence specifying the type of amino acids in the protein and the overall 3D structure. The protein backbone

consists of $N - C_\alpha - C - O$ atoms. Researchers often focus on the location of the $C_\alpha$ atom. Together with the torsion angles $\phi$ and $\psi$, backbone residues are denoted as rigid body frames around the $C_\alpha$ atoms. These frames have to be modeled by SE(3)-equivariant networks with a carefully designed diffusion process for the torsion angles. Although the frame representation has been popular, some works such as Genie (Lin & AlQuraishi, 2023) and Proteína (Geffner et al., 2025) only use the coordinates of the $C_\alpha$ atoms to represent the protein backbone in the diffusion and flow matching process and have demonstrated notable success in the designability of generated backbones. The benefit of modeling only the $C_\alpha$ atoms is that the backbone can simply be presented in as an $N \times 3$ matrix of Cartesian coordinates, where $N$ is the number of residues in the backbone.

**Flow matching.** Here we take rectified flow (Liu et al., 2023) as an example. It models the process of gradually transitioning from the noise distribution $p_0(\boldsymbol{x}_0) \sim \mathcal{N}(0, \mathbf{I})$ to the data distribution $p_1(\boldsymbol{x}_1)$. This probability density path $p_t(\boldsymbol{x}_t)$ for $t \in [0, 1]$ is modeled as an ordinary differential equation (ODE): $d\boldsymbol{x}_t = \boldsymbol{v}_t^\phi(\boldsymbol{x}_t, t)$, where $\boldsymbol{v}_t^\phi(\boldsymbol{x}_t, t)$ is the velocity estimated by a the neural network. To avoid confusion, we will denote real data as $\boldsymbol{x}_d$ and noise as $\boldsymbol{\epsilon}$ in all subsequent derivations. During training, each step on the probability path is defined as

$$\boldsymbol{x}_t = t\boldsymbol{x}_d + (1 - t)\boldsymbol{\epsilon}_t \tag{1}$$

and the target velocity to approximate is

$$\boldsymbol{v}_t^\phi(\boldsymbol{x}_t, t) \approx \boldsymbol{x}_d - \boldsymbol{\epsilon}_t, \tag{2}$$

where $\boldsymbol{x}_d$ represents real data and $\boldsymbol{\epsilon}_t \sim \mathcal{N}(0, \mathbf{I})$.

This is similar to the forward process of diffusion models with $\boldsymbol{x}_t = a_t\boldsymbol{x}_d + \sigma_t\boldsymbol{\epsilon}_t$, where $a_t = t$ and $\sigma_t = 1 - t$. Indeed, as mentioned in Albergo et al. (2023) and Geffner et al. (2025), flow matching models and diffusion models using Gaussian noise can be shown to be equivalent upon reparametrization. Hence, the same score matching ideas in SiD can be applied to flow matching models with certain adjustments, as we discuss later.

**Diffusion distillation with SiD.** In general, diffusion models involve the forward process and the reverse process, adding and removing noise at each step respectively. During the forward process, clean data $\boldsymbol{x}_d$ is gradually transformed into random noise $\boldsymbol{\epsilon}$ step by step. For Gaussian diffusion, at each step denoted as time $t$, the noised data can be written as $\boldsymbol{x}_t = a_t\boldsymbol{x}_d + \sigma_t\boldsymbol{\epsilon}_t$, where the noise schedule $a_t$ and $\sigma_t$ are predefined. The score function can be written as $\nabla_{\boldsymbol{x}_t}q(\boldsymbol{x}_t|\boldsymbol{x}_d) = \frac{1}{\sigma_t^2}(a_t\boldsymbol{x}_d - \boldsymbol{x}_t)$. Given a pretrained model with parameters $\phi$, denoted as $f_\phi$, we define $f_\phi(\boldsymbol{x}_t, t)$ as its real data prediction, or its approximation to $\mathbb{E}(\boldsymbol{x}_d|\boldsymbol{x}_t, t)$. Then we can write its score as $S_\phi(\boldsymbol{x}_t) = \frac{1}{\sigma_t^2}(a_t f_\phi(\boldsymbol{x}_t, t) - \boldsymbol{x}_t)$. In SiD, this score is assumed to be a good approximation to the score of data distribution $\nabla_{\boldsymbol{x}_t}\ln p_{\text{data}}(\boldsymbol{x}_t) \approx S_\phi(\boldsymbol{x}_t)$ at all timesteps $t$. The goal is to match the distribution of the one-step generator with parameter $\theta$, denoted as $G_\theta$, to the data distribution by indirectly matching the generator score to the score of the pretrained model $f_\phi$. To achieve this, another score network with parameters $\psi$, the generator score network $f_\psi$, is trained to approximate the score of the generator, and the final target becomes: to match the generator score network $f_\psi$ and $f_\phi$ using their scores $S_\psi(\boldsymbol{x}_t)$ and $S_\phi(\boldsymbol{x}_t)$ at all timesteps $t$.

**Related Work in Protein Backbone Generation.** Two seminal works in protein backbone generation are Chroma (Ingraham et al., 2023), which uses a correlated diffusion process for the protein backbones, and RFDiffusion (Watson et al., 2023), which relies on pretraining from the protein structure prediction model RosettaFold (Baek et al., 2021). It utilizes the RosettaFold frame representation and adds Gaussian noise and Brownian motion to the translation and the orientation parts in the backbone frames respectively in its diffusion process. FrameDiff (Yim et al., 2023b) followed the frame representation and applied SE(3) invariant diffusion. FrameFlow (Yim et al., 2023a) and FoldFlow (Bose et al., 2024) build on the idea but uses flow matching as a faster alternative. Proteus (Wang et al., 2024) introduced a more efficient triangle layer with the help of graph-based methods. While these frame-based methods have performed really well in generating new protein backbone structures, SMCDiff (Trippe et al., 2023), Genie (Lin & AlQuraishi, 2023), its subsequent Genie2 (Lin et al., 2024), and the recent Proteína (Geffner et al., 2025) have shown that modeling only the coordinates of the alpha carbon atoms can produce promising results.

**Related Work in Diffusion Distillation.** A major limitation in diffusion and flow-based models is the sampling speed, as they usually require hundreds and sometimes up to a thousand sampling

steps to reliably move from a known noise distribution (e.g. the standard Gaussian distribution) to the unknown data distribution. Significant efforts have been made to improve the sampling speed by reducing the number of sampling steps while still modeling the path from the noise distribution to the data distribution. One of the first ideas is progressive distillation (Salimans & Ho, 2022), which halves the number of steps after each iteration. Another important work is consistency models (Song et al., 2023) which learns the ODE trajectory so that the model can consistently denoise to clean samples from anywhere along the ODE trajectory. Flow Map Matching (Boffi et al., 2025) and Consistency Trajectory Models (Kim et al., 2024) extends the idea by not only modeling the flow map from a point in the trajectory to the clean data, but also the flow map between any two points in the trajectory. Align-Your-Flow builds on the flow map matching idea and demonstrated promising results in few-step image generation. In the meantime, significant efforts have been put into distribution matching, pioneered by Diffusion-GAN (Wang et al., 2023a). The idea is to match the distributions of the generator $p_\theta(\boldsymbol{x}_t)$ and the data distribution $p_{data}(\boldsymbol{x}_t)$ at all timesteps $t$ in the forward diffusion process. Wang et al. (2023a) minimized the Jensen-Shannon divergence between the two distributions. Other works like Wang et al. (2023b) and Luo et al. (2024) use the KL divergence instead. In more recent works Zhou et al. (2024) and Zhou et al. (2025b) considered the Fisher divergence as an alternative. An advantage of using the Fisher divergence is that it can be solely based on the score function learned by the teacher model to facilitate data-free distillation. When real data are available, they can also be incorporated into the distillation scheme as shown in Zhou et al. (2025b) and Zhou et al. (2025a). To our knowledge, none of these distillation strategies have been successfully applied to protein backbones, likely due to the structural sensitivity of protein structures. Indeed, our initial attempts to adapt existing score distillation methods to this setting were largely unsuccessful.

## 3 METHOD

Both diffusion- and flow-based models have been popular and successful in the protein backbone generation literature. However, applying a image distillation method directly often leads to catastrophic failures, since protein backbone generation models often require low temperature sampling to boost designability. As one-step generators are not capable of noise scaling that is crucial for stable protein generation, we propose a few-step distillation + noise scaling approach. In Section 3.1, we first show that SiD can be extended to a distillation strategy for both flow matching and diffusion by deriving the equivalent flow matching distillation scheme for protein structures. We focus on the rectified flow formulation (Liu et al., 2023) as an example of the class of flow matching models and use the $\mathcal{M}_{\text{FS}}^{\text{no-tri}}$ model in Proteína as the pretrained teacher model as it achieves high designability and low sampling time. In Section 3.2, we extend the one-step distillation technique to few-step distillation. In Section 3.3, we incorporate the noise scaling factor in our few-step sampling. Lastly, in Section 3.4, we discuss some other important hyperparameters that ensure stable generation.

### 3.1 SiD FOR FLOW MATCHING

As discussed in Albergo et al. (2023), Geffner et al. (2025), and various other papers, flow matching and diffusion models are equivalent upon reparametrization and the key quantity linking the two classes of models is the score function, which is also the core of SiD (Zhou et al., 2024). SiD estimates the score of the data distribution as $\nabla_{\boldsymbol{x}_t}\ln p_{data}(\boldsymbol{x}_t) \approx S_\phi(\boldsymbol{x}_t) = \sigma_t^{-2}(f_\phi(\boldsymbol{x}_t, t) - \boldsymbol{x}_t)$, where $f_\phi(\boldsymbol{x}_t, t) \approx \mathbb{E}(\boldsymbol{x}_d|\boldsymbol{x}_t)$ is the predicted true data $\boldsymbol{x}_d$ by the pretrained model conditioned on the noised data $\boldsymbol{x}_t$ at timestep $t$. The target of SiD is to minimize the score difference between the pretrained network $f_\phi$ and the generator network $G_\theta$ for all $t$:

$$\min_\theta \mathbb{E}_{\boldsymbol{x}_t \sim p_\theta(\boldsymbol{x}_t)}[\|S_\phi(\boldsymbol{x}_t) - \nabla_{\boldsymbol{x}_t}\ln p_\theta(\boldsymbol{x}_t)\|_2^2] \quad (3)$$

In order to apply SiD to a flow matching model, we first derive the score function for rectified flow as $\nabla_{\boldsymbol{x}_t}\ln p_{data}(\boldsymbol{x}_t) \approx (1-t)^{-2}(tf_\phi(\boldsymbol{x}_t, t) - \boldsymbol{x}_t)$ where $f_\phi(\boldsymbol{x}_t, t) = \boldsymbol{x}_t + (1-t)v_t^\phi(\boldsymbol{x}_t, t)$. To estimate the score of the generator in Eq. 3, we train another neural network $f_\psi$ using the usual flow matching loss. We treat the generator output $\boldsymbol{x}_g$ as clean data and define the fake score training objective:

$$\min_\psi \mathbb{E}_{q(\boldsymbol{x}_t|\boldsymbol{x}_g, t)p_\theta(\boldsymbol{x}_g)}\left[\frac{1}{N(1-t)^2}\|f_\psi(\boldsymbol{x}_t, t) - \boldsymbol{x}_g\|_2^2\right] \quad (4)$$

where $N$ is the number of residues.

Following the formulation in Zhou et al. (2024), we derive the initial approximation to the target score difference in Eq. 3 as:

$$\mathcal{L}_\theta^{(1)} = \mathbb{E}_{\boldsymbol{x}_t \sim p_\theta(\boldsymbol{x}_t)}[\|\frac{t}{(1-t)^2}(f_\phi(\boldsymbol{x}_t, t) - f_\psi(\boldsymbol{x}_t, t)\|_2^2] \tag{5}$$

and the projected score matching loss as:

$$\mathcal{L}_\theta^{(2)} = \frac{t^2}{(1-t)^4}\mathbb{E}_{\boldsymbol{x}_t \sim p_\theta(\boldsymbol{x}_t), \boldsymbol{x}_t \sim q(\boldsymbol{x}_t \mid \boldsymbol{x}_g, t)}[(f_\phi(\boldsymbol{x}_t, t) - f_\psi(\boldsymbol{x}_t, t))^T(f_\phi(\boldsymbol{x}_t, t) - \boldsymbol{x}_g)] \tag{6}$$

As discussed in Zhou et al. (2024), minimizing $\mathcal{L}_\theta^{(1)}$ alone may not lead to meaningful results, and a fused loss of both $\mathcal{L}_\theta^{(1)}$ and $\mathcal{L}_\theta^{(2)}$ in the form of $\mathcal{L}_\theta = \mathcal{L}_\theta^{(2)} - \alpha\mathcal{L}_\theta^{(1)}$ needs to be considered. The authors of SiD (Zhou et al., 2024) reported that $\alpha \in [0.75, 1.2]$ yielded the best results. We have observed similar behavior in our ablation study and decided to set $\alpha = 1.0$ in all our experiments (see Appendix C).

With derivations detailed in Appendix D, the final generator loss takes the form:

$$\mathcal{L}_\theta = \frac{\omega(t)t^2}{(1-t)^4}[(1-\alpha)\|f_\phi(\boldsymbol{x}_t, t) - f_\psi(\boldsymbol{x}_t, t)\|_2^2 + (f_\phi(\boldsymbol{x}_t, t) - f_\psi(\boldsymbol{x}_t, t))^T(f_\psi(\boldsymbol{x}_t, t) - \boldsymbol{x}_g)] \tag{7}$$

And the generator loss reweighting function $\omega(t)$ is defined as:

$$\omega(t) = \frac{(1-t)^4}{Nt^2\|\boldsymbol{x}_g - f_\phi(\boldsymbol{x}_g)\|_{1,\text{sg}}} \tag{8}$$

We defer the detailed derivation of the relationship between the velocity $\boldsymbol{v}_t^\phi(\boldsymbol{x}_t, t)$ and the neural network output $f_\phi(\boldsymbol{x}_t, t)$ and why we can use the neural network output directly in our losses to Appendix D.

## 3.2 FEW-STEP GENERATION

Although one step generators trained by SiD have been shown to perform well in image generation, we have observed that our one step protein generator produces almost no designable samples despite extensive efforts to finetune the training process. Adding guidance, switching to other distillation strategies such as Diff-Instruct (Luo et al., 2024), trying different loss reweighting, and tuning hyperparameters all failed to improve the performance of the one-step generator in any significant way. Our hypothesis is that since small errors in local structures can render a backbone completely undesignable, the common practice of low temperature sampling needs to be incorporated in our distilled models as well. There is no corresponding "trick" that can be applied in the one-step distillation scheme for such low temperature sampling. The only operation that might be related is scaling down the standard Gaussian noise at the start of generation, which did not prove helpful. We also attempted to add guidance to the generator but the designability remained extremely low. Therefore, to improve the performance and to allow low temperature sampling, we modify the SiD Few-Step framework (Zhou et al., 2025a) to distill few-step generators. In the few-step version of the score-based distillation scheme, samples are generated in the following way as specified in Zhou et al. (2025a) and Song et al. (2023):

$$\boldsymbol{x}_g^{(k)} = G_\theta(t_k\text{sg}(\boldsymbol{x}_g^{(k-1)}) + (1 - t_k\boldsymbol{\epsilon}_k), \quad \boldsymbol{\epsilon}_k \sim \mathcal{N}(0, \mathbf{I}), \tag{9}$$

where $\gamma$ is the noise scaling factor, $t_k$ is the $k$th step in a predefined time path $t_{steps}$, $\boldsymbol{x}_g^{(k)}$ is the generated sample at step $k$, and sg stands for the stop gradient function, which is important during training since we use the uniform-step matching approach specified in Zhou et al. (2025a).

During training, we uniformly sample a step $k \in \{1, ..., K\}$ where $K$ is the target number of steps for our generator and generate $\boldsymbol{x}_g^{(k-1)}$ using Eq. 9. In the last step of generating $\boldsymbol{x}_g^{(k)}$ from $\boldsymbol{x}_g^{(k-1)}$, we do not apply the stop gradient operation to allow for gradient tracking. This way, we avoid the need for back propagation through the entire generation chain, which would be limited by GPU memory in practice.

### 3.3 INFERENCE TIME NOISE SCALING

Our few-step generators distilled from the strategy above still have extremely low designability using the standard sampling procedure. Without low temperature sampling, the distillation strategy matches the generator's distribution to the distribution learned by the pretrained model $f_\phi$, which may not suffice because the pretrained model itself is not exactly generating from its learned distribution. Now that we have more than 1 sampling steps, a similar noise scale can be added at each sampling step for our few-step generators. We modify the sampling step of Eq. 9 as:

$$\boldsymbol{x}_g^{(k)} = G_\theta(t_k \boldsymbol{x}_g^{(k-1)} + \gamma(1 - t_k)\boldsymbol{\epsilon}_k), \quad \boldsymbol{\epsilon}_k \sim \mathcal{N}(0, \mathbf{I}), \tag{10}$$

where $\gamma$ is the noise scaling factor. The choice of the noise scale $\gamma$ greatly affects the generation quality as discussed in Appendix B. We decided to set $\gamma = 0.45$, the same value as used in Proteína, which gives the optimal balance between designability and diversity.

The sampling steps without noise scaling as defined in Eq. 9 is used when generating samples during distillation training. On the other hand, the sampling steps with noise scaling as defined in Eq. 10 is applied when producing actual protein structures for evaluation.

### 3.4 ADDITIONAL TRAINING DETAILS

**Number of Residues.** Reweighting both the generator loss and the fake score loss by $N$, the number of residues in the protein structure, is crucial so that the generator can learn to generate proteins of different lengths. As a completely data-free distillation scheme, we do not require any existing databases like the PDB (Berman et al., 2000) or AFDB (Varadi et al., 2021). We uniformly sample the lengths within the supported range (in all our experiments the range is set to 50-256) in each training batch and use the mask for loss reweighting. While we are data-free, there is potential to further enhance our method by utilizing these high quality data by adding the adversarial components, we leave that for future study.

**Time schedule.** During training, we follow the log schedule in Proteína when sampling the timestep $t$. Choosing $p = 2$ and $N_{steps} = 400$, we first sample $s \sim \text{Unif}[0, N_{steps}]$, then define $t$ as:

$$t = 1 - 10^{-\frac{s}{N_{step}}p} \tag{11}$$

The time steps for our generator is defined as equally spaced points between $T_{init}$ and $N_{steps}$, before they are converted to continuous time using Eq. 11. $T_{init}$ is set to 30 which corresponds to a signal-to-noise-ratio of around 1/2.5, in order to match the $\sigma_{init}$ value of 2.5 in the original SiD paper (Zhou et al., 2024).

**Fold Class Conditioning.** For evaluation, we also test our distilled generators' ability to generate structures conditioned on the fold class labels. We follow the same definition for the fold class labels defined in Proteína. In short, the fold classes are based on the CATH structural hierarchy (Dawson et al., 2016) which describes different levels of the protein structure and the CAT level labels are used for the fold classes.

**Parameters for SiD.** The ablation study for $\alpha$ and other parameter settings are discussed in Appendix C. The detailed training procedure is presented in Algo. 1 and sampling procedure in Algo. 2.

## 4 EXPERIMENTS

To evaluate our distillation framework, we trained our one-step and few-step generators using the pretrained model from Proteína, specifically the $\mathcal{M}_{\text{FS}}^{\text{no-tri}}$ model which is the fastest in terms of sampling time. We then evaluated the distilled generators with different sampling steps against the pretrained model for the tasks of unconditional protein structure generation and fold class conditional generation. Furthermore, we performed biological plausibility analyses to verify that the generated protein structures are indeed biologically meaningful.

### 4.1 EVALUATION

We adopt the evaluation pipeline from Proteína, which assesses designability, diversity, and novelty through a self-consistency test involving ProteinMPNN (Dauparas et al., 2022) and ESMFold (Lin

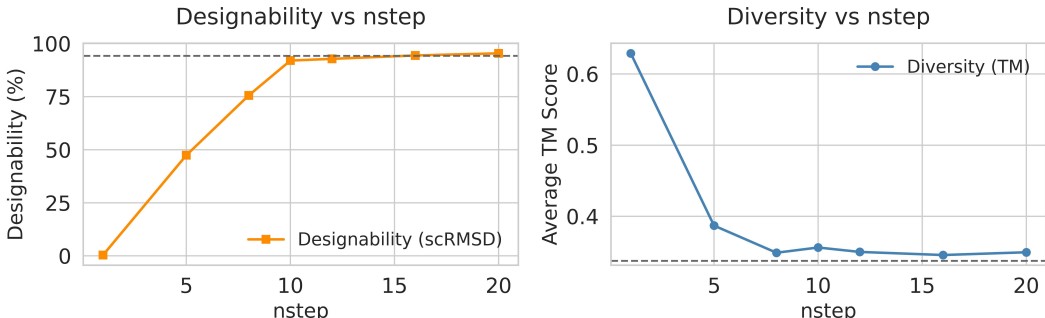

Figure 1: Plots of designability and diversity versus the number of generation steps. On the left, we show the designability as the percentage of generated samples that meet the designable threshold (scRMSD < 2). On the right, we show the diversity as the average pairwise TM-scores between designable samples. The lower the average TM-score is, the more diverse the generated samples are. 16 steps seems enough to beat the pretrained model in designability, while being slightly worse in diversity.

et al., 2023). Notably, in this pipeline, a structure is considered designable if the best self-consistency root mean squared deviation (scRMSD) is less than 2Å. The final designability is then calculated based on the percentage of all generated samples that are designable. The details of the metrics are discussed in the Proteína paper (Geffner et al., 2025). For each generator that we distilled, we generate 100 samples for each length in $\{50, 100, 150, 200, 250\}$ and calculate the evaluation metrics for designability, diversity, and novelty, as well as the metrics proposed in Proteína (FPSD, fS, and fJSD). In addition to the metrics for the quality of generation, we also record the total time taken to generate all 500 samples for sampling time analysis. See Appendix E for details about the implementation of the evaluation pipeline.

## 4.2 UNCONDITIONAL GENERATION IN ONE STEP

As mentioned, we first applied our flow-matching-adjusted score distillation to train an one-step generator. As expected, the one-step generator produced very few designable structures, as shown in the first row of Tab. 1. Attempts to improve the performance by adding autoguidance and classifier-free guidance (CFG), modifying the SiD losses, and tuning training parameters all failed to produce any meaningful improvements. Therefore, it is crucial to switch to multi-step generators to enable low temperature sampling, as our results confirmed the ineffectiveness of one-step distillation for proteins.

## 4.3 UNCONDITIONAL GENERATION IN MULTIPLE STEPS

Multistep + low temperature sampling improves designability drastically. Since the SiD Few-Step framework is flexible in terms of the number of steps, we trained generators with a range of numbers of steps and compared their performance. We find that just introducing the sampling procedure defined in Eq. 9 is not enough to bring notable improvements to the designability of the distilled generators, regardless of the number of sampling steps. However, adding the noise scale to the sampling procedure as described in Eq. 10 significantly improves the generation quality.

To further reduce the sampling time, we trained generators with $K \in \{16, 12, 10, 8, 5\}$ steps using the SiD Few-Step algorithm and compared their generation quality. We also reran the pretrained $\mathcal{M}_{\mathrm{FS}}^{\mathrm{no\text{-}tri}}$ model to ensure a fair comparison. The results are presented in Fig. 1. Evidently, as the number of sampling steps increases from 1, both the designability and the diversity improves steadily. Compared to the pretrained model, all our few-step generators produced more diverse samples, but the level of designability of the pretrained model is only reached when we increase our number of sampling steps to more than 10.

Table 1: Unconditional backbone generation performance for our distilled generators with different sampling steps compared to the Proteína pretrained model $\mathcal{M}_{FS}^{no\text{-}tri}$ and the distilled generators using Diff-Instruct.

| Number of Steps $K$ | Design-ability (%)↑ | Diversity Cluster↑ | TM↓ | Novelty PDB↓ | Novelty AFDB↓ | FPSD PDB↓ | FPSD AFDB↓ | fS (C / A / T)↑ | fJSD PDB↓ | fJSD AFDB↓ | Sec. Struct. % ($\alpha/\beta$) | Sampling Time (s) |
|---|---|---|---|---|---|---|---|---|---|---|---|---|
| 1 | 0.4 | 1.00 (2) | 0.63 | 0.78 | 0.80 | 2470.17 | 2403.07 | 2.31 / 2.37 / 4.45 | 2.74 | 2.27 | 75.1 / 0.1 | 5.687 |
| 5 | 47.4 | 0.64 (151) | 0.39 | 0.81 | 0.81 | 457.29 | 423.92 | 1.04 / 2.05 / 11.74 | 3.48 | 2.54 | 71.5 / 0.0 | 0.172 |
| 8 | 75.6 | 0.77 (292) | 0.35 | 0.76 | 0.77 | 525.78 | 524.53 | 1.20 / 2.43 / 14.75 | 3.06 | 2.25 | 66.9 / 0.7 | 0.167 |
| 10 | 92.0 | 0.60 (277) | 0.36 | 0.78 | 0.79 | 586.74 | 586.73 | 1.20 / 2.74 / 10.69 | 3.16 | 2.38 | 71.2 / 0.8 | 0.169 |
| 12 | 92.8 | 0.65 (300) | 0.35 | 0.79 | 0.79 | 558.26 | 547.21 | 1.24 / 2.87 / 10.96 | 3.05 | 2.27 | 69.6 / 1.1 | 0.200 |
| 16 | 94.4 | 0.61 (287) | 0.35 | 0.80 | 0.81 | 404.04 | 394.20 | 1.86 / 4.27 / 16.18 | 1.94 | 1.54 | 63.2 / 4.6 | 0.261 |
| 20 | 95.4 | 0.60 (286) | 0.35 | 0.80 | 0.81 | 515.35 | 510.83 | 1.44 / 3.29 / 13.79 | 2.66 | 2.01 | 68.2 / 2.0 | 0.320 |
| 16 ($\gamma = 1$) | 41.4 | 0.79 (164) | 0.33 | 0.79 | 0.80 | 142.2 | 146.0 | 2.51 / 5.33 / 35.46 | 0.62 | 0.48 | 44.9 / 9.0 | 0.710 |
| 10-step Diff-Instruct | 0.74 | 0.92 (34) | 0.35 | 0.76 | 0.78 | 729.93 | 678.33 | 1.26 / 1.96 / 10.42 | 3.21 | 2.44 | 64.0 / 1.0 | 0.213 |
| $\mathcal{M}_{FS}^{no\text{-}tri}$ | 94.2 | 0.63 (297) | 0.34 | 0.83 | 0.84 | 316.73 | 290.66 | 1.92 / 4.78 / 20.57 | 1.80 | 1.24 | 64.4 / 4.6 | 6.395 |

Tab. 1 shows evaluation metrics for our generators of various numbers of steps, compared with a 10-step generator trained with Diff-Instruct (with similar modification to the distillation training and low temperature sampling) and the pretrained model. Our 16- and 20-step generators can achieve higher designability than the pretrained model, while maintaining a comparable diversity. Our distilled generators are also better at generating novel structures different from those in the reference databases. The alpha helix and beta sheet contents in the generated structures are about the same between our 16-step generator and the pretrained model. Moreover, we can reach even higher designability by increasing the number of steps to 20, and potentially more. However, for the three metrics proposed by Geffner et al. (2025), namely FPSD, fS, and fJSD, our generators performed worse than the pretrained model in all categories.

As shown in Tab. 2, our sampled structures from the distilled model did not appear to under- or over-represent a certain fold class. The "failure" rate, defined as the ratio between the number of undesignable structures and the total number of structures belonging to each class, did not demonstrate any notable difference compared to the pretrained model. This confirms that our distillation process is fair and did not favor a particular type of proteins.

Table 2: "Failure" rate within each fold class at the "C" level. The labels are specific to the Fold Classifier provided by Proteína (Geffner et al., 2025), not the actual "C" codes. The number in the parenthesis represents the number of samples predicted to be within each class.

| Fold class label | 0 | 1 | 2 | 3 | 4 |
|---|---|---|---|---|---|
| Distilled | 0.039 (358) | 0.200 (5) | 0.098 (132) | NA (0) | 5 |
| Pretrained | 0.036 (366) | 0.181 (11) | 0.087 (115) | 0 (1) | 0.571 (7) |

Our hypothesis on why these fold-class-related metrics are worse for our distilled model is that they naturally correlate with the diversity and novelty of our generated samples. As our diversity metrics are indeed slightly worse than the pretrained model, it is reasonable to assume that there are less diverse fold-classes, leading to lower fS scores. Similarly, as suggested by the novelty scores, our distilled model is able to produce more novel proteins, which may lead to reduced distributional similarity with the reference databases. This would naturally result in worse FPSD and fJSD scores.

Finally, we show that our distilled model is still capable of full distribution sampling (by setting the noise scale $\gamma$ to 1). It more closely matches the full distribution as evident in the reduced FPSD and fJSD scores.

## 4.4 SAMPLING TIME

Here we highlight the improvements in the sampling time for our generators with different numbers of steps. Although sampling time strictly decreases with fewer generation steps, it is crucial to account for the quality of the generated structures. Therefore, we measure the effective sampling time by dividing the total time taken to generate all 500 samples, in the unconditional generation experiment, by the number of designable samples. The last column of Tab. 1 shows the results for the effective sampling time using an A6000-48GB GPU in batches of size 10. All our generators achieved better effective sampling time than the pretrained model. The one-step generator has a significantly larger effective sampling time due to its near-zero designability. All other generators displayed an improvement from the pretrained model by more than 20 times. Fig. 5 in Appendix F provides an illustration of the impact of the number of steps on the effective sampling time. It

Table 3: Fold class-conditional generation performance for our distilled generator compared to the Proteína pretrained model $\mathcal{M}_{\text{FS}}^{\text{no-tri}}$.

| Model | Design-ability (%)↑ | Diversity Cluster↑ | TM↓ | Novelty PDB↓ | AFDB↓ | FPSD PDB↓ | AFDB↓ | fS (C / A / T)↑ | fJSD PDB↓ | AFDB↓ | Sec. Struct. % $(\alpha/\beta)$ |
|---|---|---|---|---|---|---|---|---|---|---|---|
| 16-step distilled $\mathcal{M}_{\text{FS}}^{\text{no-tri}}$ | 95.0 | 0.78 (74) | 0.35 | 0.82 | 0.83 | 473.21 | 456.62 | 1.96 / 4.87 / 13.01 | 1.94 | 1.69 | 61.8 / 5.2 |
| pretrained $\mathcal{M}_{\text{FS}}^{\text{no-tri}}$ | 96.0 | 0.82 (79) | 0.34 | 0.83 | 0.84 | 428.82 | 395.73 | 1.88 / 4.29 / 19.01 | 2.01 | 1.43 | 65.0 / 4.2 |

confirms that although fewer number of steps generally lead to reduced effective sampling time, lowering the number of steps to fewer than 10 is not beneficial as the designability of the generated samplings decrease drastically.

## 4.5 FOLD CLASS CONDITIONAL GENERATION

We also show that our distilled models are capable of fold class-conditional generation. With the 16-step generator distilled unconditionally from the pretrained model, we provide the fold class labels as input and evaluate the conditionally generated structures. The labels are sampled from the empirical joint distribution of length and CATH (Dawson et al., 2016) code, obtained from proteins in AFDB (Varadi et al., 2021). As shown in Tab. 3, our distilled generator can achieve comparable metrics as the pretrained model in fold class-conditional generation, although a similar drop in the fold-class-related metrics is observed.

## 4.6 BIOLOGICAL PLAUSIBILITY ANALYSES

To evaluate the structural and functional plausibility of our designed proteins, we conducted a case study on a representative design with high composite confidence (plddt = 0.936, ptm = 0.896, RMSD = 0.545), with high novelty score (seqid = 0.14), confirming that the generated sequence is distinct from any known proteins. Figure 2a shows the protein backbone, which features a stable helical bundle with well-packed secondary structure. Importantly, automated cavity detection (Fpocket) revealed the presence of two adjacent surface pockets (Figure 2b): one predominantly polar and the other largely hydrophobic.

Pocket 1 (red) has a druggability score of 0.758, a volume of 596 Å$^3$, and a polarity score of 7 (polar-leaning), while Pocket 2 (orange) has a druggability score of 0.796, a volume of 507 Å$^3$, and a polarity score of 3 (hydrophobic-leaning). These scores suggests both pockets are potential pharmaceutically relevant binding sites, with sufficient space to accommodate small molecules and complementary chemical environments. The combination of adjacent chemically distinct pockets supports two plausible binding modes: 1) dual-ligand occupancy, with a polar ligand in Pocket 1 and a hydrophobic ligand in Pocket 2, potentially allowing cooperative binding; 2) single bifunctional ligand, where a fragment-linked molecule bridges both pockets, exploiting complementary polar and nonpolar interactions.

The novel structure of chemically distinct binding sites within a compact fold highlights the potential of our generative pipeline to produce proteins with functionally relevant ligand-binding opportunities. Importantly, this observation underscores not only the structural realism of the designs but also their direct usability in downstream tasks such as molecular recognition and drug discovery.

## 4.7 APPLICATION TO DIFFUSION MODEL WITH EQUIVARIANT ARCHITECTURE

We have shown that our distillation framework can successfully distill a few-step generator from a pretrained Proteína model, which is flow-based and does not have equivariant architecture or IPA and triangle layers. To verify that our distillation framework can work on diffusion-based models and that it can handle equivariant networks with IPA and triangle layers, we distilled a 10-step generator from the pretrained 1000-step Genie model (Lin & AlQuraishi, 2023). Note that the generator loss in SiD requires calculating the gradient with respect to $x_g$ which is the input to the fake score network and the teacher network. However, Genie2 contains a one-hot-encoding operation which would cut off the gradient with respect to the input of the neural network. Therefore, we cannot directly apply SiD to Genie 2 (Lin et al., 2024) without first resolving the gradient backpropagation issue introduced by its one-hot encoding operation. We additionally report the designability based

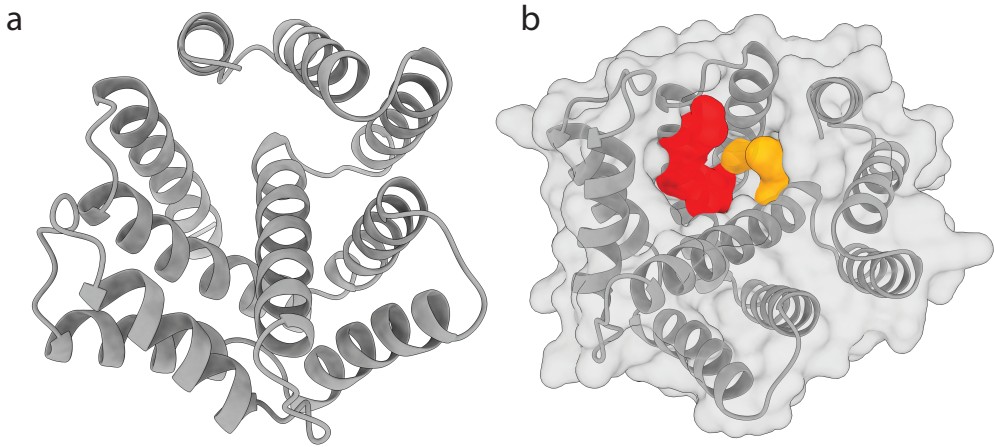

Figure 2: Designed protein backbone and adjacent polar–hydrophobic pocket pair. (a) Front view of the designed protein backbone. (b) Cavity view highlighting two adjacent but chemically distinct pockets: Pocket 1 (red, polar) and Pocket 2 (orange, hydrophobic). The complementary polarities enable two use modes: simultaneous binding of a polar and a hydrophobic ligand, or one bifunctional molecule spanning both pockets.

on scTM which calculates the TM score instead of the RMSD in the self-consistency pipeline and uses a threshold of 0.5 as described in Lin & AlQuraishi (2023).

The result shows that our distillation framework can also be applied to diffusion models based on equivariant architecture with IPA and triangle layers, achieving 100-fold speedup in generation while actually improving the designability of the samples. The

Table 4: Unconditional generation performance for our distilled generator compared to the Genie pretrained model.

| Model | Designability (%) | | Diversity | | Novelty | |
|---|---|---|---|---|---|---|
| | scTM ↑ | scRMSD ↑ | Cluster↑ | TM↓ | PDB↓ | AFDB↓ |
| 10-step distilled Genie | 87.0 | 57.0 | 0.66 (45) | 0.38 | 0.79 | 0.79 |
| 1000-step pretrained Genie | 85.0 | 0.39 | 0.85 (33) | 0.35 | 0.78 | 0.80 |

diversity of our distilled model is worse than the pretrained model however. We think it is likely due to suboptimal choice of the noise scale and expect a more thorough investigation would be needed in order to find the noise scale that achieves the best balance between designability and diversity, which is the standard practice in protein structure generation literature.

## 5 CONCLUSION

We have adapted a state-of-the-art diffusion distillation strategy for image diffusion models, SiD, for diffusion and flow-based Protein backbone generative models. We extend its capabilities beyond one-step distillation to few-step generation with support for low temperature sampling, a critical practice in protein backbone generation models. We demonstrate effectiveness of our new few-step distillation technique by distilling the flow-based Proteína model with entirely data-free training. Experimental results in unconditional and fold class-conditional generation show that the distilled multistep generators lead to a more than 20-fold reduction in effective sampling time, while achieving comparable performance to the teacher model in terms of the key metrics of designability, diversity, and novelty. Higher designability can be achieved by increasing the number of sampling steps, as demonstrated by our 20-step generator. We hypothesize that the modest drop in the fold-class-related metrics can be resolved if we relax the data-free condition and actually sample the empirical joint distribution of length and fold class. We also verify the biological plausibility through a case study of a generated structure. Overall, this result takes us one step closer to large-scale protein design, and we hope to find ways to accelerate other components of the design pipeline, especially the folding model, such as AlphaFold3 (Abramson et al., 2024).

## REPRODUCIBILITY STATEMENT

We ensure full reproducibility of our training, sampling, and evaluation pipeline. The model architecture and weights of the pretrained Proteína model (Geffner et al., 2025) are publicly available. The distillation code is based on SiD (Zhou et al., 2024), and our evaluation pipeline integrates publicly available components from Genie (Lin & AlQuraishi, 2023) and Proteína (Geffner et al., 2025). Other models used for evaluation such as FoldSeek (van Kempen et al., 2024), ProteinMPNN (Dauparas et al., 2022), ESMFold (Lin et al., 2023), and TM (Zhang & Skolnick, 2004) are publicly available as well. Therefore, our training, sampling, and evaluation pipeline are fully reproducible.

Our code will be available on request after the initial review and during the rebuttal period. The weights of the distilled models will be made publicly available, subject to the original license constraint of Proteína .

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

APPENDIX

## A   ALGORITHM

Algorithm 1 details the training procedure for our distillation.

Algorithm 2 specifies the inference-time sampling procedure. Note that at inference, Eq. 10 with noise scale is used instead of the standard Eq. 9 used during training.

---

**Algorithm 1** SiD one- or few-step training loop for protein backbone generation

---

1: **Input:** Pretrained score network $f_\phi$, generator $G_\theta$, generator score network $f_\psi$, $t_{init} = 0.37$, $t_{min} = 0.02$, $t_{max} = 0.98$, number of generation steps $K = 16$, $\alpha = 1.0$, optimizer learning rates $\eta_\psi = 1e^{-4}$ and $\eta_\theta = 5e^{-5}$.

2: **Initialization:** $\theta \leftarrow \phi, \psi \leftarrow \phi, t_{steps} \leftarrow K$ equally spaced points between 30 and 400

3: **repeat**

4:     Sample target protein length $N \in \{50, \cdots, 256\}$.

5:     **if** The number of generation steps $K = 1$ **then**

6:         Sample $z \sim \mathcal{N}(0, \mathbf{I})$ and set $x_g = G_\theta(t_{\text{init}}z)$

7:     **else**

8:         Sample $k$ from $\{1, ..., K\}$, then sample $x_g$ in $k$ steps recursively using Eq. (9)

9:     **end if**

10:    Sample $t$ using Eq. (11), and clamp $t$ to be between $t_{min}$ and $t_{max}$.

11:    Sample $\epsilon_t \sim \mathcal{N}(0, \mathbf{I})$. Set $x_t = t\text{sg}(x_g) + (1 - t)\epsilon_t$

12:    Update $\psi$ using 4:

13:        $\mathcal{L}_\psi = \frac{1}{N(1-t)^2}\|f_\psi(x_t, t) - \text{sg}(x_g)\|_2^2$, where $N$ is the number of residues.

14:        $\psi \leftarrow \psi - \eta_\psi \nabla_\psi \mathcal{L}_\psi$

15:    **if** The number of generation steps $K = 1$ **then**

16:        Sample $z \sim \mathcal{N}(0, \mathbf{I})$ and set $x_g = G_\theta(t_{\text{init}}z)$

17:    **else**

18:        Sample $k$ from $\{1, ..., K\}$ sample $x_g$ in $k$ steps using Eq. 9

19:    **end if**

20:    Sample $t$ using Eq. (11), and clamp $t$ to be between $t_{min}$ and $t_{max}$. Compute $\omega(t)$ as defined in Eq. (8).

21:    Sample $\epsilon_t \sim \mathcal{N}(0, \mathbf{I})$. Set $x_t = tx_g + (1 - t)\epsilon_t$

22:    Update $\theta$ using Eq. (7)

23:        $\mathcal{L}_\theta = \frac{\omega(t)t^2}{(1-t)^4}[(1-\alpha)\|f_\phi(x_t, t) - f_\psi(x_t, t)\|_2^2 + (f_\phi(x_t, t) - f_\psi(x_t, t))^T(f_\psi(x_t, t) - x_g)]$

24:        $\theta \leftarrow \theta - \eta_\theta \nabla_\theta \mathcal{L}_\theta$

25: **until** The average scRMSD plateaus or the training budget is exhausted

26: **Output:** $G_\theta$

---

---

**Algorithm 2** SiD one- or few-step inference for protein backbone generation

---

1: **Input:** Generator $G_\theta$, $t_{init} = 0.37$, $\gamma = 0.45$, number of generation steps $K = 16$, target protein length $N$.

2: **Initialization:** $t_{steps} \leftarrow K$ equally spaced points between 30 and 400

3: **if** The number of generation steps $K = 1$ **then**

4:     Sample $z \sim \mathcal{N}(0, \mathbf{I})$ and set $x_g = G_\theta(t_{\text{init}}z)$

5: **else**

6:     Sample $x_0 \sim \mathcal{N}(0, \mathbf{I})$

7:     **for** $k \in \{1, \cdots, K\}$ **do**

8:         Sample $x_{k+1}$ from $x_k$ using Eq. (10) with noise scale $\gamma$ and time path $t_{steps}$

9:     **end for**

10: **end if**

11: **Output:** $x_K$

---

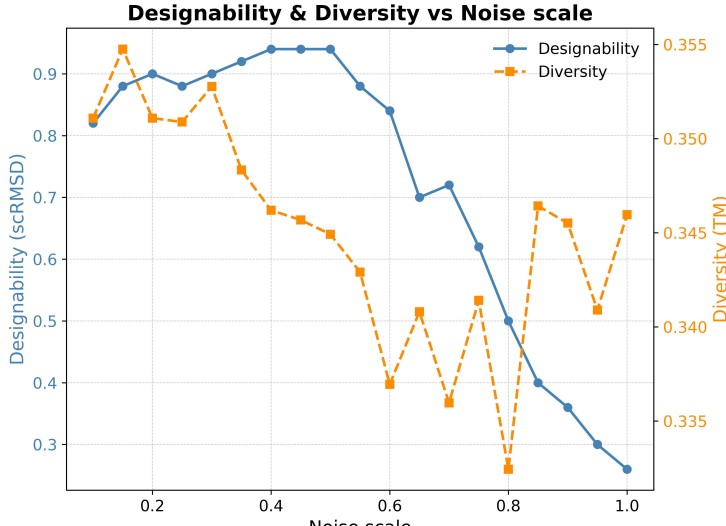

Figure 3: An illustration of the effects of the noise scale on the designability (scRMSD) and diversity (TM) of our 10-step generator. For designability, it is the fraction of samples in the batch with scRMSD < 2Å. The higher the designability the better. For the diversity metric, it is measuring the average similarity (TM-score) within the batch. Lower values indicate more diversity. Setting the noise scale to 1 is the standard way of sampling in diffusion and flow matching models for images, which results in close-to-0 designability for protein structures. The designability is the highest for noise scales around 0.45 while the best diversity is reached at 0.8.

## B  EFFECT OF SAMPLING NOISE SCALE

As discussed by prior works, the noise scale during sampling is crucial for protein structure generation tasks. We empirically verify the impact of noise scaling and determine our optimal noise scale.

As shown in Fig. 3, the designability is the highest when the noise scale is around 0.45. It drastically drops as the noise scale increases beyond 0.5. As for the diversity, it worsens significantly as the noise scale decreases below 0.4. Therefore, it seems the optimal noise scale for our 10-step generator is between 0.4 and 0.5. We chose 0.45 eventually, which coincides with the value picked for the $\mathcal{M}_{\text{FS}}^{\text{no-tri}}$ model in Proteína .

## C  ABLATION STUDY AND PARAMETER SETTINGS

**Impact of $\alpha$.** Similar to the original SiD paper (Zhou et al., 2024), we conduct an ablation study for the impact of $\alpha$ on the distillation training process. Designability metrics including the average scTM, the average scRMSD, and the designability based on the proportions of structures with scRMSD < 2Å in a batch are plotted in Fig. 4 as it evolves from 0 to 102.4 thousand samples processed during training. Across the $\alpha$ values of [0, 0.5, 0.8, 1.0, 1.2, 1.5], we see the best designability performance with $\alpha$ being set to 0.8 or 1.0. With other $\alpha$ values, we do not observe a meaningful convergence towards an improved designability. Between the values 0.8 and 1.0, it seems setting $\alpha = 1.0$ results in a slightly smoother and more stable convergence. Hence, we select $\alpha = 1$ for all our experiments.

**Impact of $\beta_1$.** We follow a similar experimental process as in the SiD paper (Zhou et al., 2024) to investigate the impact of the $\beta_1$ parameter for the Adam optimizer. We compare the performance of setting $\beta_1$ to 0 and 0.9 respectively for the optimizer for the generator $G_\theta$. We did not observe any notable difference in either the convergence speed or the final designability. The $\beta_1$ parameter for the optimizer for the fake score network, $f_\psi$ was kept at 0. As reported in Zhou et al. (2024), setting $\beta_1 = 0.9$ often does not result in convergence.

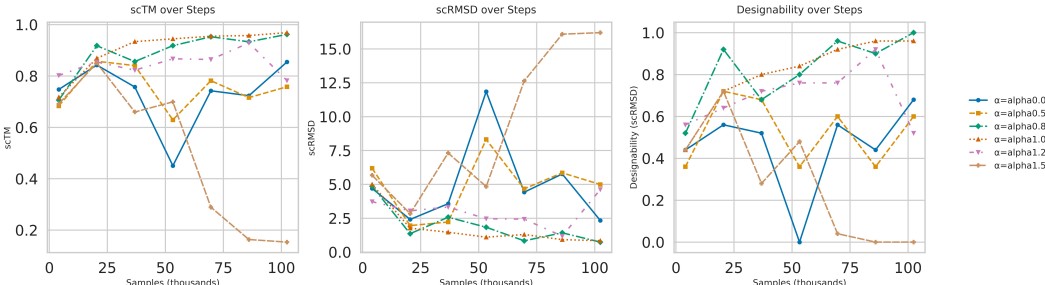

Figure 4: Ablation study of $\alpha$. Each plot illustrates the generation quality, measured in the average scTM, average scRMSD, and scRMSD-based designability within each generated batch against the number of training samples processed during distillation across different $\alpha$ values. The ablation study showcases the effect of $\alpha$ on distillation training and highlights the reason why $\alpha$ is set to 1.0 for subsequent experiments.

**Impact of batch size and learning rate.** We initially set the overall batch size to 256 and failed to observe any noticeable improvement in designability. As we increase the batch size to 1024, 2048 and eventually 4096, we observe more and more stable convergence to the level of designability of the pretrained model, with the combination of batch size set to 4096 and learning rate set to 1e-4 and 5e-5 for $f_\psi$ and $G_\theta$ respectively resulting in our best performance. Although it is tempting to conclude that the larger the batch size the better, it is also possible that the unstable performance was due to the learning rate not being optimized for the smaller batch sizes. A more comprehensive grid search might be needed to uncover the optimal setting for the batch size and learning rate. It is also worth noting that the batch size per GPU was set to 2 because the model can be memory-intensive for longer proteins. This results in a large number of gradient accumulation rounds.

## D    PROOFS

As the score function is the key quantity for SiD, we derive it under the flow matching formulation in Proteína . For a pretrained network $f_\phi$ with original data $\boldsymbol{x}_d$ and noised sample $\boldsymbol{x}_t$ at timestep $t$, the score can be written as:

$$S_\phi(\boldsymbol{x}_t) = \frac{t\boldsymbol{v}_t^\phi(\boldsymbol{x}_t, t) - \boldsymbol{x}_t}{1 - t} \tag{12}$$

$$= \frac{t\mathbb{E}(\boldsymbol{x}_d - \boldsymbol{\epsilon}_t | \boldsymbol{x}_t) - \boldsymbol{x}_t}{1 - t} \tag{13}$$

$$= \frac{\mathbb{E}(t\boldsymbol{x}_d - t\boldsymbol{\epsilon}_t - (t\boldsymbol{x}_d + (1 - t)\boldsymbol{\epsilon}_t) | \boldsymbol{x}_t)}{1 - t} \tag{14}$$

$$= -\frac{\mathbb{E}(\boldsymbol{\epsilon}_t | \boldsymbol{x}_t)}{1 - t} \tag{15}$$

$$= -\frac{\mathbb{E}(\frac{1}{1-t}(\boldsymbol{x}_t - t\boldsymbol{x}_d) | \boldsymbol{x}_t)}{1 - t} \tag{16}$$

$$= (1 - t)^{-2}\mathbb{E}((t\boldsymbol{x}_d - \boldsymbol{x}_t) | \boldsymbol{x}_t) \tag{17}$$

$$= (1 - t)^{-2}(t\mathbb{E}(\boldsymbol{x}_d | \boldsymbol{x}_t) - \boldsymbol{x}_t) \tag{18}$$

$$= (1 - t)^{-2}(tf_\phi(\boldsymbol{x}_t, t) - \boldsymbol{x}_t) \tag{19}$$

where $f_\phi(\boldsymbol{x}_t, t) = \boldsymbol{x}_t + (1 - t)v_t^\phi(\boldsymbol{x}_t, t)$ and $v_t^\phi$ is the velocity function for $f_\phi$.

The score difference objective in Eq. 3 also involves the score of the generator, $\nabla_{\boldsymbol{x}_t}\ln p_\theta(\boldsymbol{x}_t)$. To estimate the generator score, we use another neural network $f_\psi$ and train it using the same training objective as Proteína .

$$\min_\psi \mathbb{E}_{q(\boldsymbol{x}_t | \boldsymbol{x}_g, t)p_\theta(\boldsymbol{x}_g)}[\frac{1}{N}\|\boldsymbol{v}_t^\psi(\boldsymbol{x}_t, t) - (\boldsymbol{x}_g - \boldsymbol{\epsilon})\|_2^2] \tag{20}$$

, where $N$ denotes the number of residues in the desired protein structure.

We can derive the relationship between the network estimation $f_\phi(\boldsymbol{x}_t, t)$ and the velocity function $v_t^\phi(\boldsymbol{x}_t, t)$:

$$f_\phi(\boldsymbol{x}_t, t) = \mathbb{E}(\boldsymbol{x}_d | \boldsymbol{x}_t) \tag{21}$$

$$= \mathbb{E}(\frac{1}{t}(\boldsymbol{x}_t - (1-t)\boldsymbol{\epsilon}_t | \boldsymbol{x}_t) \tag{22}$$

$$= \frac{1}{t}(\boldsymbol{x}_t - (1-t)\mathbb{E}(\boldsymbol{x}_d - \boldsymbol{v}_t^\phi(\boldsymbol{x}_t, t) | \boldsymbol{x}_t) \tag{23}$$

$$\Rightarrow t f_\phi(\boldsymbol{x}_t, t) = \boldsymbol{x}_t - (1-t)f_\phi(\boldsymbol{x}_t, t) + (1-t)v_t^\theta(\boldsymbol{x}_t, t) \tag{24}$$

$$\Rightarrow f_\phi(\boldsymbol{x}_t, t) = \boldsymbol{x}_t + (1-t)\boldsymbol{v}_t^\phi(\boldsymbol{x}_t, t) \tag{25}$$

$$\Rightarrow \boldsymbol{v}_t^\phi(\boldsymbol{x}_t, t) = \frac{1}{1-t}(f_\phi(\boldsymbol{x}_t, t) - \boldsymbol{x}_t) \tag{26}$$

and therefore,

$$\boldsymbol{v}_t^\phi(\boldsymbol{x}_t, t) - (\boldsymbol{x}_d - \boldsymbol{\epsilon}) = \frac{1}{1-t}(f_\phi(\boldsymbol{x}_t, t) - \boldsymbol{x}_t) - \boldsymbol{x}_d + \boldsymbol{\epsilon} \tag{27}$$

$$= \frac{1}{1-t}f_\phi(\boldsymbol{x}_t, t) - \frac{1}{1-t}(t\boldsymbol{x}_d + (1-t)\boldsymbol{\epsilon}) - \boldsymbol{x}_d + \boldsymbol{\epsilon} \tag{28}$$

$$= \frac{1}{1-t}f_\phi(\boldsymbol{x}_t, t) - (\frac{t}{1-t} + 1)\boldsymbol{x}_d \tag{29}$$

$$= \frac{1}{1-t}(f_\phi(\boldsymbol{x}_t, t) - \boldsymbol{x}_d) \tag{30}$$

Note that the network $f_\psi$ is essentially trained by treating the samples $\boldsymbol{x}_g$ produced by the generator $G_\theta$ as the true clean data. Thus, the above relationship can be rewritten for $f_\psi$ as $\boldsymbol{v}_t^\psi(\boldsymbol{x}_t, t) - (\boldsymbol{x}_g - \boldsymbol{\epsilon}) = \frac{1}{1-t}(f_\psi(\boldsymbol{x}_t, t) - \boldsymbol{x}_g)$ and the score of $f_\psi$ is $S_\psi(\boldsymbol{x}_t) = (1-t)^{-2}(t\mathbb{E}(\boldsymbol{x}_g \mid \boldsymbol{x}_t) - \boldsymbol{x}_t)$.

Hence, Eq. 20 is equivalent to

$$\min_\psi \mathbb{E}_{q(\boldsymbol{x}_t | \boldsymbol{x}_g, t)p_\theta(\boldsymbol{x}_g)}[\frac{1}{N(1-t)^2}\|f_\psi(\boldsymbol{x}_t, t) - \boldsymbol{x}_g\|_2^2] \tag{31}$$

We can then approximate the loss in Eq. 3 as the score difference between $f_\phi$ and $f_\psi$, and define:

$$\mathcal{L}_\theta^{(1)} = \mathbb{E}_{\boldsymbol{x}_t \sim p_\theta(\boldsymbol{x}_t)}[\|S_\phi(\boldsymbol{x}_t) - S_\psi(\boldsymbol{x}_t)\|_2^2] \tag{32}$$

$$= \mathbb{E}_{\boldsymbol{x}_t \sim p_\theta(\boldsymbol{x}_t)}[\|\frac{t}{(1-t)^2}(f_\phi(\boldsymbol{x}_t, t) - f_\psi(\boldsymbol{x}_t, t)\|_2^2] \tag{33}$$

Expanding the $L_2$ norm of the target score difference in Eq. 3, we have:

$$\mathbb{E}_{\boldsymbol{x}_t \sim p_\theta(\boldsymbol{x}_t)}[\|S_\phi(\boldsymbol{x}_t) - \nabla_{\boldsymbol{x}_t}\ln p_\theta(\boldsymbol{x}_t)\|_2^2] \tag{34}$$

$$\approx \mathbb{E}_{\boldsymbol{x}_t \sim p_\theta(\boldsymbol{x}_t)}[(S_\phi(\boldsymbol{x}_t) - S_\psi(\boldsymbol{x}_t))^T(S_\phi(\boldsymbol{x}_t) - \nabla_{\boldsymbol{x}_t}\ln p_\theta(\boldsymbol{x}_t))] \tag{35}$$

$$= \mathbb{E}_{\boldsymbol{x}_t \sim p_\theta(\boldsymbol{x}_t)}[(S_\phi(\boldsymbol{x}_t) - S_\psi(\boldsymbol{x}_t))^T(\frac{1}{(1-t)^2}(t\mathbb{E}(\boldsymbol{x}_d | \boldsymbol{x}_t) - \boldsymbol{x}_t))] \tag{36}$$

$$- \mathbb{E}_{\boldsymbol{x}_t \sim p_\theta(\boldsymbol{x}_t)}\mathbb{E}_{\boldsymbol{x}_t \sim q(\boldsymbol{x}_t \mid \boldsymbol{x}_g, t)}[(S_\phi(\boldsymbol{x}_t) - S_\psi(\boldsymbol{x}_t))^T\nabla_{\boldsymbol{x}_t}\ln q(\boldsymbol{x}_t \mid \boldsymbol{x}_g)] \tag{37}$$

$$= \mathbb{E}_{\boldsymbol{x}_t \sim p_\theta(\boldsymbol{x}_t)}[(S_\phi(\boldsymbol{x}_t) - S_\psi(\boldsymbol{x}_t))^T(\frac{1}{(1-t)^2}(t\mathbb{E}(\boldsymbol{x}_d | \boldsymbol{x}_t) - \boldsymbol{x}_t))] \tag{38}$$

$$- \mathbb{E}_{\boldsymbol{x}_t \sim p_\theta(\boldsymbol{x}_t)}\mathbb{E}_{\boldsymbol{x}_t \sim q(\boldsymbol{x}_t \mid \boldsymbol{x}_g, t)}[(S_\phi(\boldsymbol{x}_t) - S_\psi(\boldsymbol{x}_t))^T(\frac{1}{(1-t)^2}(t\boldsymbol{x}_g - \boldsymbol{x}_t))] \tag{39}$$

$$= \frac{t}{(1-t)^2}\mathbb{E}_{\boldsymbol{x}_t \sim p_\theta(\boldsymbol{x}_t)}\mathbb{E}_{\boldsymbol{x}_t \sim q(\boldsymbol{x}_t \mid \boldsymbol{x}_g, t)}[(S_\phi(\boldsymbol{x}_t) - S_\psi(\boldsymbol{x}_t))^T(\mathbb{E}(\boldsymbol{x}_d | \boldsymbol{x}_t) - \boldsymbol{x}_g)] \tag{40}$$

Therefore, we define

$$\mathcal{L}_\theta^{(2)} = \frac{t}{(1-t)^2} \mathbb{E}_{\boldsymbol{x}_t \sim p_\theta(\boldsymbol{x}_t), \boldsymbol{x}_t \sim q(\boldsymbol{x}_t \mid \boldsymbol{x}_g, t)}[(S_\phi(\boldsymbol{x}_t) - S_\psi(\boldsymbol{x}_t))^T (f_\phi(\boldsymbol{x}_t, t) - \boldsymbol{x}_g)] \tag{41}$$

$$= \frac{t}{(1-t)^2} \mathbb{E}_{\boldsymbol{x}_t \sim p_\theta(\boldsymbol{x}_t), \boldsymbol{x}_t \sim q(\boldsymbol{x}_t \mid \boldsymbol{x}_g, t)}\left[\frac{t}{(1-t)^2}(f_\phi(\boldsymbol{x}_t, t) - f_\psi(\boldsymbol{x}_t, t))^T (f_\phi(\boldsymbol{x}_t, t) - \boldsymbol{x}_g)\right] \tag{42}$$

$$= \frac{t^2}{(1-t)^4} \mathbb{E}_{\boldsymbol{x}_t \sim p_\theta(\boldsymbol{x}_t), \boldsymbol{x}_t \sim q(\boldsymbol{x}_t \mid \boldsymbol{x}_g, t)}[(f_\phi(\boldsymbol{x}_t, t) - f_\psi(\boldsymbol{x}_t, t))^T (f_\phi(\boldsymbol{x}_t, t) - \boldsymbol{x}_g)] \tag{43}$$

and thus the final SiD loss, $\mathcal{L}_\theta = \mathcal{L}_\theta^{(2)} - \alpha\mathcal{L}_\theta^{(1)}$ as:

$$\mathcal{L}_\theta = \frac{\omega(t)t^2}{(1-t)^4}[(1-\alpha)\|f_\phi(\boldsymbol{x}_t, t) - f_\psi(\boldsymbol{x}_t, t)\|_2^2 + (f_\phi(\boldsymbol{x}_t, t) - f_\psi(\boldsymbol{x}_t, t))^T (f_\psi(\boldsymbol{x}_t, t) - \boldsymbol{x}_g)] \tag{44}$$

## E   EVALUATION PIPELINE

Our evaluation pipeline largely follows that described in Geffner et al. (2025), with the same metrics for designability, diversity, and novelty, as well as its proposed metrics of FPSD, fS, and fJSD. In the self-consistency pipeline for **designability**, each generated structure is passed to ProteinMPNN (Dauparas et al., 2022) to generate 8 candidate sequences. Each of the 8 sequences is fed to ESMFold (Lin et al., 2023) for predict the corresponding structure. Each ESMFold-predicted structure is then compared with the original generated structure in terms of RMSD. The best RMSD among the 8 predicted structures is reported as the scRMSD. A generated structure is considered designable if the scRMSD is less than 2Å. **Diversity** is measured by both the average pairwise TM-score (Zhang & Skolnick, 2004) among designable samples and the proportion of designable clusters among all designable samples. **Novelty** is calculated by first computing the highest TM-score between each generated structure and each structure in the reference dataset, either PDB (Berman et al., 2000) or AFDB (Varadi et al., 2021). The average of these maximum TM-scores is reported as the novelty metric. The **secondary structure content** is calculated based on the P-SEA algorithm (Labesse et al., 1997). The implementation of the evaluation pipeline, however, is based on both Proteína (Geffner et al., 2025) and Genie (Lin & AlQuraishi, 2023). We used the code for the evaluation pipeline provided by Genie to calculate the scRMSD, diversity based on the TM-score, and the secondary structure contents. For the cluster-based diversity and the novelty metrics, we followed the Foldseek commands specified in Proteína . In addition, we use the implementation and the fold class predictor network provided by Proteína for the FPSD, fS, and fJSD metrics. Since the implementation is slightly different, we reran the metrics on the pretrained model instead of copying over the reported results to ensure a fair comparison. For unconditional generation, we generate 100 samples with batch size 10 for protein lengths [50, 100, 150, 200, 250]. We use an Nvidia A6000-48GB GPU and track the total time for generating all samples for timing analyses.

## F   IMPACT OF NUMBER OF STEPS ON EFFECTIVE SAMPLING TIME

Although the intuition is that the generation time strictly decreases as the number of sampling steps decrease, it is important to account for the quality of the generated structures in the evaluation of sampling time. As shown in Fig. 5, our one-step generator, on average, takes the longest time by far to generate one designable sample compared to other generators. The reason is that the drop in designability performance outweighs the gain in generation speed for the one-step generator. We argue that the effective sampling time more faithfully reflects the speed of generation, compared to simple metrics such as the number of sampling steps or time taken to generate one sample.

## G   THE USE OF LARGE LANGUAGE MODELS (LLM)

LLMs were used to check grammar and spelling and polish wording.

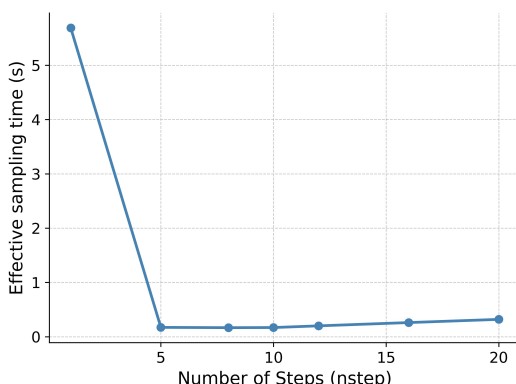

Figure 5: Effective Sampling Time against the number of generator steps. 500 batches generated in batches of 10 on an A6000-48GB GPU.

## H    EXAMPLE OF GENERATED STRUCTURES

## I    EXAMPLE OF FAILURE CASES

Examining these undesignable structures carefully, it is evident that our one-step generator favors alpha helices. No beta sheet has been generated in these samples. Additionally, there exists some exceptionally long alpha helices, such as the ones in the last example of the $N = 100$ row and the first sample in the $N = 150$ row. Furthermore, the alpha helices tend to appear in 180 degree angles to one another, which is again unusual in stable protein structures. Overall, the one-step generator was not able to model the geometric properties of protein structures.

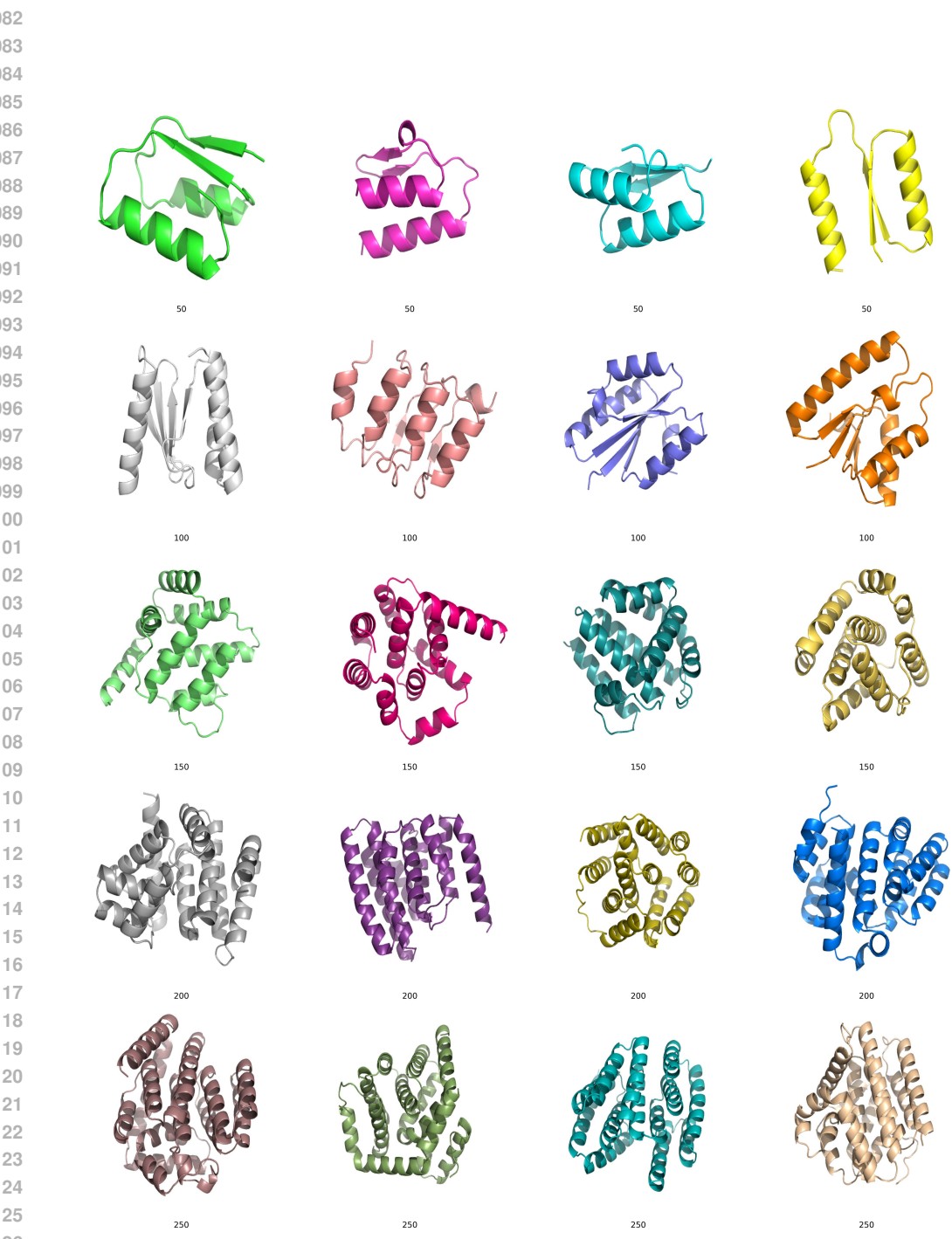

Figure 6: Example of structures generated unconditionally of varying lengths. The lengths are displayed below each structure. All structures shown here are designable.

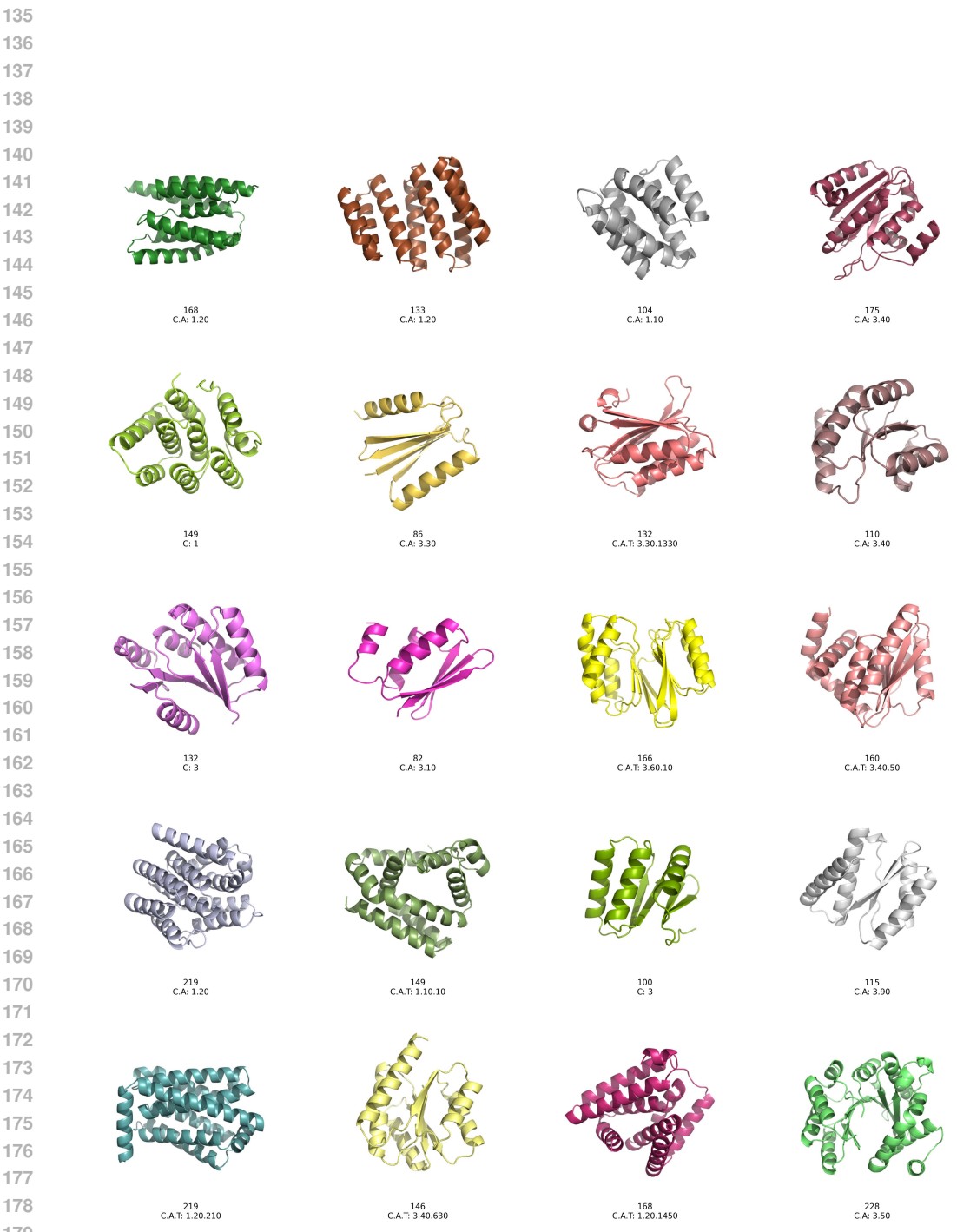

Figure 7: Examples of generated structures conditioned on the fold class as specified by the CATH code. The length and the C.A.T fold class label are denoted below the structures. All structures shown here are designable.

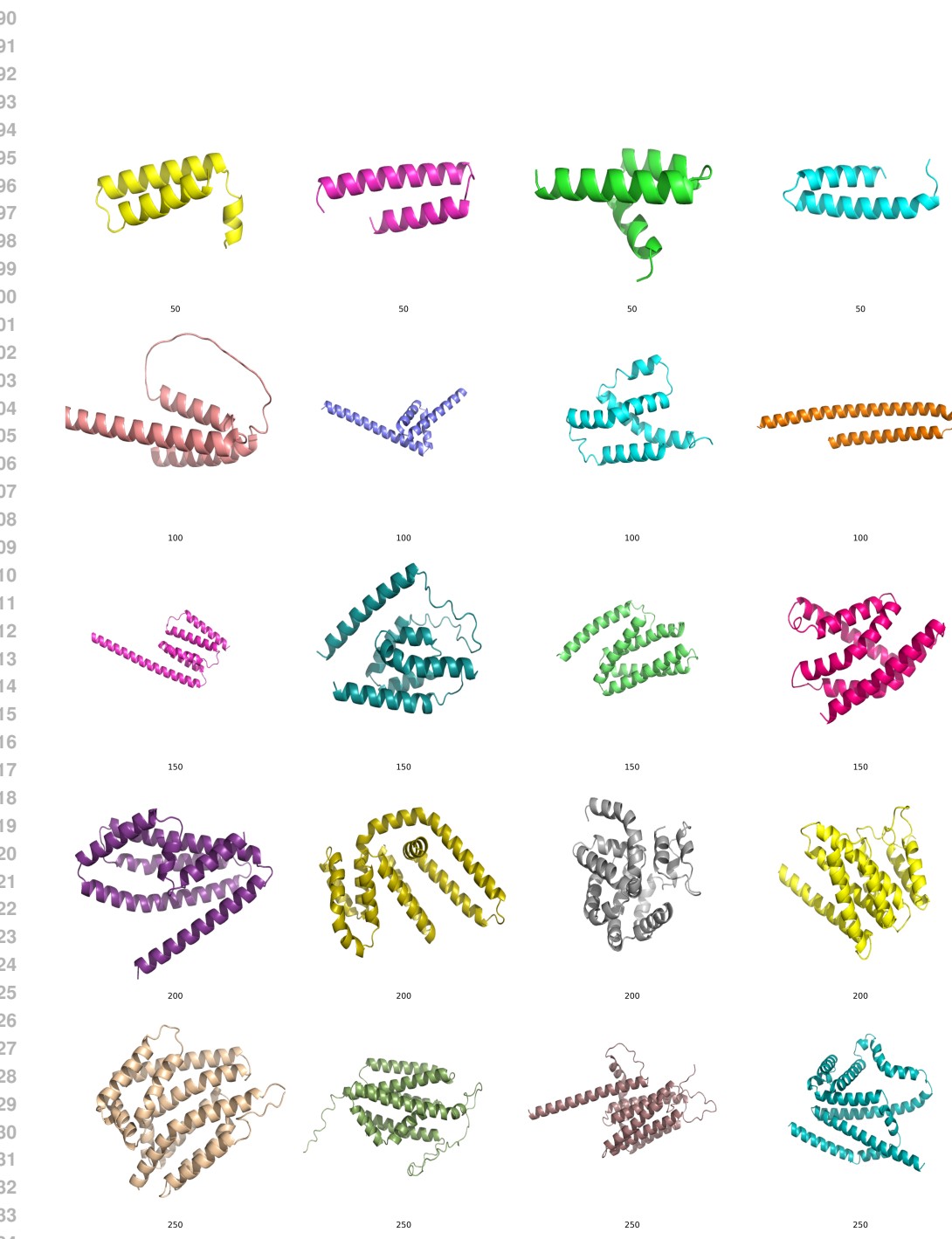

Figure 8: Examples of structures generated unconditionally by our one-step distilled generator that are considered undesignable. The lengths are displayed below each structure.