# OpenReview forum: "Distilled Protein Backbone Generation"
_ICLR.cc/2026/Conference — Submitted to ICLR 2026_

### Official Review · Reviewer_bFFC · 2025-10-27

**Soundness:** 3
**Presentation:** 2
**Contribution:** 3
**Rating:** 4
**Confidence:** 5

**Summary:**

This paper presents a method for accelerating flow-based protein backbone generation models by applying a score distillation technique, specifically Score Identity Distillation (SiD). The authors adapt SiD, originally from the vision domain, to distill a pre-trained protein generator (Proteína) into a few-step model. The key challenge addressed is successfully combining the distillation process with the low-temperature sampling (inference-time noise scaling) required to maintain high designability in protein structures. The resulting distilled model achieves a significant (e.g., >20x) speedup while maintaining comparable designability, diversity, and novelty to the original teacher model.

**Strengths:**

The primary strength of this work is the successful application of an acceleration technique to the challenging domain of protein generation. While the method (SiD) is not novel, its adaptation is non-trivial. The authors demonstrate how to make score distillation compatible with the low-temperature sampling crucial for protein designability, which is a valuable practical contribution. The paper shows promising results, achieving a significant speedup while preserving the key quality metrics of the generated backbones.

**Weaknesses:**

1. **Weak Motivation**: The central motivation—that generation speed is a practical bottleneck for large-scale protein discovery—is not well-supported. The paper claims this limits discovery pipelines needing thousands of samples. However, even the "slow" teacher model (e.g., ~1 hour for 600 samples, based on the paper's reported times) is already orders of magnitude faster than the subsequent wet-lab validation. This makes the 20x speedup less impactful for this specific use case. The motivation would be stronger if framed in a different context, such as large-scale in silico screening or other inference-time scaling scenarios where generation speed is the true bottleneck.
2. **Limited Novelty & Incomplete Related Work**: The work is an application of an existing method (SiD). This is acceptable, but the paper lacks a comprehensive overview of other recent advances in few-step diffusion generation (e.g., MeanFlow, Shortcut Models, Align your Flow, Flow Map Matching). The authors should discuss these alternatives and provide a clear justification for why SiD is the most suitable choice for this specific task over other potential distillation or fast-sampling methods.
3. **Clarity of Presentation**: The core algorithms for training and inference are relegated to the appendix. For clarity and reproducibility, it would be much better to include algorithm boxes in the main paper to describe the full training and inference procedures.

**Questions:**

In Eq (4), what is the intuition behind the hyperparameter $\alpha$? While Appendix C provides an ablation study, the main text should briefly explain its role in balancing the loss terms and how it was set in practice.

---

> ### Author Response · Authors · 2025-11-21
> **Response to Reviewer bFFC - Part 1**
>
> We thank Reviewer bFFC for their comprehensive review. We appreciate that the reviewer considers our adaptation of the distillation method to the protein generation domain “non-trivial” and “is a valuable practical contribution”. Below we address the raised questions.
>
> 1. *Bottleneck of large-scale protein discovery*
>
> **Response:** We would like to point out that modern drug and protein discovery pipelines routinely rely on high-throughput experimental screening (HTS). In large pharmaceutical settings, plate-based HTS can assay on the order of 10^5–10^6 small molecules within a few days [1], and more specialized technologies—such as droplet microfluidics, ultra-HT functional screens, and display platforms can effectively explore diversity in the 10^7 to even 10^10+ range in pooled formats [2]. On the other hand, the hit rate of HTS ranges between 0.01% and 0.1%. Most of the screened compounds are routinely reported as inactive towards the desired bioactivity [3]. Due to the high throughput and low hit rate, experimental screening requires a large pool of candidates.
> In diffusion-based protein design, the practical workflow consists of (1) in silico generation of large candidate sets, (2) in silico filtering using structure prediction and energy models, and (3) wet-lab screening of the surviving designs. Because the vast majority of designs fail at the in silico stage (effective failure rates often >90–99%), only a small fraction (≪10%, sometimes around 1%) of generated candidates can actually reach the wet-lab test phase. Therefore, even a moderately high-throughput experimental campaign that tests 10^4 designed proteins may require on the order of 10^6 candidates, and standard HTS screening requires more than 10^8 candidates.
>
> Current diffusion model can generate hundreds of samples per hour, making sampling 10^6 candidates already costs hundreds to thousands of computational hours for a single design round, not to mention one needs 10^8 candidates for the HTS setting. This makes sampling speed of diffusion-based protein structure generators a critical bottleneck in scaling model-driven design to match industrial high-throughput screening regimes.
>
> We do recognize that with improved speed in generating large candidate sets, in silico screening involving folding models remains a critical bottleneck. We plan to explore methods to make folding models more efficient in future projects.
>
> 2. *Comparison with recent advances in few-step diffusion generation*
>
> **Response:** We thank the reviewer for highlighting this point. We did compare against another data-free distillation technique, Diff-Instruct [4], and showed in Table 1 that at the same number of sampling steps, SiD achieves substantially higher designability. A similar quality gap between SiD and Diff-Instruct has also been observed in image generation benchmarks. Notably, Diff-Instruct is a one-step KL-based method, while SiD minimizes Fisher divergence. To adapt Diff-Instruct for few-step distillation, we adopted the SiD framework and changed the optimization target. Among data-free one/few-step distillation methods, SiD currently represents the state of the art in image domains, motivating our choice to extend it to protein generation.
>
> Regarding other recent few-step or one-step approaches, we note a key distinction between diffusion pretraining + distillation methods and few-step generators trained from scratch. Methods such as Shortcut Models [5] and MeanFlow [6] fall into the latter category; they do not rely on a pretrained diffusion model and are thus less relevant to our setup, which focuses on distilling large pretrained teachers. Moreover, their reported results (e.g., FID scores on CIFAR-10 and ImageNet) still clearly lag behind pretrained-diffusion + distillation methods like SiD. For example, for one-step generation, the FID on CIFAR10 is 2.92 for MeanFlow, while 1.92 for SiD [7] and 1.50 for adversarial SiD (SiDA) [8]. Shortcut Models achieved an FID of 10.60 on ImageNet256, while MeanFlow achieved 3.43 on the same dataset. In comparison, SiD reached 1.89 FID on ImageNet512, and SiDA further improved this to 1.37.
>
> Another important distinction is data access. SiD is fully data-free beyond the pretrained teacher, whereas Align Your Flow (AYF) [9] and Flow Map Matching [10] require access to the original training data. AYF’s code is not yet publicly available, but once it is, we encourage future work to explore combining AYF with our noise-rescaled few-step recipe to test whether comparable or improved performance can be achieved.

---

> ### Author Response · Authors · 2025-11-21
> **Response to Reviewer bFFC - Part 2**
>
> 3. *Clarity of Presentation*
>
> **Response:** We thank the reviewer for the suggestion. Due to the page limit, we are unable to move the algorithm boxes for training and inference to our main text. To address the reviewer’s concern, we have added details of our training losses and the role of the hyperparameter $\alpha$ to Section 3.1 and the important stop gradient operations during training to Section 3.2. Other important hyperparameters such as the noise scale and the time schedule have been discussed in Sections 3.3 and 3.4.
>
> Q1. *Role of alpha*
>
> **Response:** The hyperparameter alpha is native to the distillation framework of SiD. In the original SiD paper, section 4.2 describes in detail why the naïve score matching loss might fail and sections 4.3 derives the projected score matching loss, and section 4.4 demonstrates the importance of the fused loss combining both the projected score matching loss and the $\alpha$-weighted naïve score matching loss. The optimal alpha term is then decided based on empirical results. We followed a similar empirical process and identified that $\alpha=1$ gives the best results, as shown in our Appendix C and Figure 4. We have added the form of the fused loss where alpha is first introduced and discussed the optimal range and our final choice of alpha. \
> \
> References:
>
> [1] Vladisavljević GT. (2024). Droplet Microfluidics for High-Throughput Screening and Directed Evolution of Biomolecules. Micromachines, 15(8):971. https://doi.org/10.3390/mi15080971
>
> [2] Ma, F., Chung, M.T., Yao, Y. et al. (2018). Efficient molecular evolution to generate enantioselective enzymes using a dual-channel microfluidic droplet screening platform. Nat Commun 9, 1030. https://doi.org/10.1038/s41467-018-03492-6
>
> [3] Neves, B. J., Braga, R. C., Melo-Filho, C. C., Moreira-Filho, J. T., Muratov, E. N., & Andrade, C. H. (2018). QSAR-Based Virtual Screening: Advances and Applications in Drug Discovery. Frontiers in pharmacology, 9, 1275. https://doi.org/10.3389/fphar.2018.01275
>
> [4] Weijian Luo, Tianyang Hu, Shifeng Zhang, Jiacheng Sun, Zhenguo Li, and Zhihua Zhang. (2024). Diff-Instruct: A universal approach for transferring knowledge from pre-trained diffusion models. Advances in Neural Information Processing Systems, 36.
>
> [5] Frans, K., Hafner, D., Levine, S., & Abbeel, P. (2025). One step diffusion via shortcut models. International Conference on Learning Representations (ICLR 2025). OpenReview.
>
> [6] Geng, Z., Deng, M., Bai, X., Kolter, J. Z., & He, K. (2025). Mean Flows for One-step Generative Modeling. arXiv preprint arXiv:2505.13447
>
> [7] Zhou, M., Zheng, H., Wang, Z., Yin, M., & Huang, H. (2024, June 6). Score identity Distillation: Exponentially Fast Distillation of Pretrained Diffusion Models for One-Step Generation. Forty-first International Conference on Machine Learning.
>
> [8] Zhou, M., Zheng, H., Gu, Y., Wang, Z., & Huang, H. (2024, October 4). Adversarial Score identity Distillation: Rapidly Surpassing the Teacher in One Step. The Thirteenth International Conference on Learning Representations.
>
> [9] Sabour, A., Fidler, S., & Kreis, K. (2025). Align Your Flow: Scaling Continuous-Time Flow Map Distillation. arXiv preprint arXiv:2506.14603
>
> [10] Boffi, N. M., Albergo, M. S., & Vanden-Eijnden, E. (2025). Flow map matching with stochastic interpolants: A mathematical framework for consistency models. arxiv preprint arxiv:2406.07507

---

> > ### Comment · Reviewer_bFFC · 2025-11-26
> >
> > Thanks for the authors' response. Most of my concerns are addressed. However, I still think the proposed method lacks novelty, as it can be seen as a simple adpation of SiD algorithm on protein structure generation. There are limited new insights in the paper. That said, the proposed method is useful in practice, as it can accelerate the generation process.
> >
> >
> > In my opinion, there is no strong reason to reject the paper despite the lack of novelty, so I increase my score to 6.

---

> > > ### Author Response · Authors · 2025-11-27
> > > **Thank you**
> > >
> > > We are glad that we were able to address most of the reviewer’s concerns and we appreciate the reviewer acknowledging the practical value of our work and adjusting the score.
> > >
> > > Regarding the question of novelty, we would like to clarify that distilling a **protein backbone generation** model presents challenges that differ substantially from the image domain. In particular, the pretrained protein generator exhibits **poor designability** under full-distribution sampling, and requires the use of **few-step distillation with noise scaling** to maintain structural fidelity. Although the adaptation builds on an established distillation framework, the **structural sensitivity of proteins and the absence of pretrained models that produce high-quality backbones without noise scaling** make the procedure non-trivial in practice.
> > >
> > > In diffusion-based image generation, a variety of guidance techniques—such as classifier guidance, auto-guidance, and classifier-free guidance—have been developed to enhance sample fidelity. In contrast, for protein generation, noise rescaling has emerged as a critical mechanism for maintaining realistic structure formation. While such fidelity-enhancing strategies may not constitute methodological novelty on their own, they provide **domain-specific empirical insights** that we believe are valuable for understanding and advancing protein generative modeling.
> > >
> > > That said, we do acknowledge that more elegant solutions are needed to alleviate the reliance on noise scaling and make one-step distillation feasible. One of the directions we would like to pursue is further **distillation from our few-step generator**, and we encourage the community to explore other potential solutions on closing the gap.

---

### Official Review · Reviewer_C2Wj · 2025-10-31

**Soundness:** 3
**Presentation:** 2
**Contribution:** 3
**Rating:** 6
**Confidence:** 5

**Summary:**

This work presents application of Score identity Distillation with the low temperature sampling needed to generate designable protein backbones, achieving a 20x reduction in inference time.

**Strengths:**

- Highly novel. First work that I am aware of that looks at distillation coupled with low temperature sampling.
- Very clear in the methods in terms of the difficulty in translating image distillation methods to proteins due to the required precision of the local structure.
- Data free scheme makes the method a lot more generalizable and not dependent on certain weighting and clusterings.
- Strong few step performance in Table 1 with 20x speed up.

**Weaknesses:**

-  The introduction is a bit misleading. Proteina does not have an equivariant architecture. Proteina also does not use IPA or need triangle layers neither of which are essential for de novo design. Proteina shows that without triangle operations nearly identical de novo accuracy can be achieved too.
- Sampling speed is important but the speed of current models is quite fast already. Speed up is important still but the claims that all existing models are too slow are overblown. From Geffner et al. Proteina can generate tens of thousands of proteins per hour. Distilled Proteina can do hundreds of thousand per hour so it is a important improvement but the introduction is a bit too overzealous.
- Eqn 3-5 could be presented a bit mroe lcearnly given its the central technical point. The Appendix is clearer but the presentation could greatly help the understanding.
- The framework was claimed to work on diffusion and flow based models while identical under gaussian prior, the work would benefit from also applying SiD ontop of Genie2 given it is the slowest model in its class.

**Questions:**

- What is the fake score loss?
- Proteina also shows that it can be trained for backbone motif scaffolding. Does this SiD framework hold for conditional tasks given all the core flowmatching aspects remain the same?
- Does this distilaltion framework work on IPA-based architectures like Genie2?

---

> ### Author Response · Authors · 2025-11-21
> **Response to Reviewer C2Wj - Part 1**
>
> We thank Reviewer C2Wj for the comprehensive review. We appreciate that the reviewer highlights the novelty of our distillation method for low temperature sampling, the challenges posed by local structure sensitivity, and the benefits of the data-free scheme. Below we address the raised questions.
>
> 1: *Clarification on Proteina architecture*
>
> **Response:** We are sorry for the confusion. We would like to clarify that the second and third paragraph of the introduction discussed the equivariant architecture and the IPA and triangle layers as a common factor limiting generation speed for many protein structure generation models. In the examples of protein backbone generation models that we listed in paragraph 2, we left out Proteina [1] precisely because it does not have equivariant architecture with IPA and triangle layers. We also mentioned briefly in paragraph 4 that Proteina showed that even without triangle layers, it can achieve state-of-the-art performance. The message we want to convey is that even for the fastest model without any of these special layers, distillation to speed things up is greatly beneficial. If the user wants to distill a pretrained model with equivariant architecture and IPA and triangle layers, distillation would be even more important. We have added explicit mentions that Proteina also does not have IPA layers and that its architecture is based on a non-equivariant transformer to avoid confusion.
>
> 2: *Importance of speedup*
>
> **Response:** Referring to Appendix C and Table 10 in the Proteina paper [1], although it is true that the smallest model of Proteina can produce over ten thousand protein backbones per hour, that level of performance is only achieved for proteins with 100 residues at maximum batch size. As the length of the proteins increases, the total runtime would increase drastically. For example, for proteins with 200 residues, the number of samples that can be generated is less than 4000, and the sampling time increases more rapidly than linear scaling as the length increases. Therefore, speeding up the generation process is still important. We also acknowledge that in the in silico protein design pipeline, the screening process involving folding models remains a critical bottleneck, and we plan to explore methods to speed up folding models in future projects.
>
> 3. *Clarity of Eq. 3-5*
>
> **Response:** We thank the reviewer for the suggestion. We have added some additional details especially regarding the loss terms. Please see Eq. 4-6 in Section 3.1 highlighted in blue.
>
> 4. *Apply SiD to Genie2*
>
> **Response:** The generator loss in SiD [2] requires calculating the gradient with respect to x_g which is the input to the fake score network and the teacher network. However, Genie2 [3]  contains a one-hot-encoding operation which would cut off the gradient with respect to the input of the neural network. Therefore, we are unable to apply SiD to Genie2. We instead tried to distill its predecessor Genie [4] which does not have one-hot-encoding.
>
> The result (see Table 4 in the revised main text) shows that our distillation framework can also be applied to diffusion models under Gaussian prior, achieving 100-fold speedup in generation while actually improving the designability of the samples. The diversity of our distilled model is worse than the pretrained model however. We think it is likely due to suboptimal choice of the noise scale and expect a more thorough investigation would be needed in order to find the noise scale that achieves the best balance between designability and diversity, which is the standard practice in protein structure generation literature.
>
> If a future version of Genie2 reverts its one-hot encoding design back to the original implementation in Genie, we are optimistic that our method could be applied to distill it. Nevertheless, Proteína remains a newer and higher-performing model compared to Genie2.
>
> Q1. *What is the fake score loss?*
>
> **Response:** In SiD, there are three networks involved: the generator that is our main target, the pretrained teacher model whose score (true score) we want to match, and the fake score network that is trained to approximate the generator’s score (fake score). During training, we first update the weights of the fake score network based on the fake score loss, which is the usual score/flow matching loss treating the generator $\boldsymbol{x}_g$ as true data, and then update the weights of the generator based on the generator loss described in Eq. 7. We briefly mentioned that the fake score network is trained using the same flow matching objective as Proteina. To make it clearer in the main text, we have added an equation (see Eq. 4) specifying the fake score loss in the method section.

---

> ### Author Response · Authors · 2025-11-21
> **Response to Reviewer C2Wj - Part 2**
>
> Q2. *motif-scaffolding*
>
> **Response:** We thank the reviewer for this insightful question. While we do not have results to share at the moment, we expect our distillation to work on motif-scaffolding tasks as the theoretical framework remains the same. And the success of Genie 2 on both unconditional generation and motif scaffolding while only training their network on motif scaffolding tasks [3] shows that there are no fundamental differences between the two tasks. Extending this framework to conditional tasks such as motif scaffolding is an exciting next step, and we plan to explore this direction in future work.
>
> Q3. *Does this distillation framework work on IPA-based architecture like Genie2?*
>
> **Response:** See the 4th point in our previous response. In short, we are unable to directly apply the distillation framework to Genie 2, but the results in Genie showed that it can work on IPA-based architecture. \
> \
> References:
>
> [1] Geffner, T., Didi, K., Zhang, Z., Reidenbach, D., Cao, Z., Yim, J., Geiger, M., Dallago, C., Kucukbenli, E., Vahdat, A., & Kreis, K. (2025). Proteina: Scaling Flow-based Protein Structure Generative Models. The Thirteenth International Conference on Learning Representations.
>
> [2] Zhou, M., Zheng, H., Wang, Z., Yin, M., & Huang, H. (2024). Score identity Distillation: Exponentially Fast Distillation of Pretrained Diffusion Models for One-Step Generation. Forty-first International Conference on Machine Learning.
>
> [3] Lin, Y., Lee, M., Zhang, Z., & AlQuraishi, M. (2024). Out of Many, One: Designing and Scaffolding Proteins at the Scale of the Structural Universe with Genie 2. arxiv preprint. arXiv:2405.15489.
>
> [4] Lin, Y., & AlQuraishi, M. (2023). Generating Novel, Designable, and Diverse Protein Structures by Equivariantly Diffusing Oriented Residue Clouds. In Proceedings of the 40th International Conference on Machine Learning (pp. 20978-21002). PMLR.

---

### Official Review · Reviewer_gk6k · 2025-10-31

**Soundness:** 3
**Presentation:** 3
**Contribution:** 3
**Rating:** 8
**Confidence:** 4

**Summary:**

The authors take a published protein structure generation method and try to adapt it for few-step distillation methods. They show that standard techniques are not successful due to low-temperature sampling adaptions that are necessary for strong performance. After iterating different design decisions, they show that with their modified SiD methodology they can reduce the sampling time by more than 20 times while still retaining similar performance.

Contributions:

[C1]  Demonstrate the challenges of adapting few-step distillation methods to generative models that use low-temperature sampling

[C2] Develop a new distillation scheme that overcomes these challenges and investigate it in detail.

**Strengths:**

[S1] Detailed Failure Analysis: instead of just presenting the working final algorithm, the authors clearly demonstrate all the things that did not work as well as modifications that were necessary to make it work, making the paper a very useful practical resource.

[S2] Empirical perfomance: the 20x sampling time acceleration with preserved performance is impressive and useful in practical applications.

**Weaknesses:**

[W1] The authors cearly describe that their fold class metrics are lower than for the pretrained model; a more detailed analysis of what the failure model for the model here is and what folds it underrepresents might be interesting.

[W2] The authors limit their evaluations only to unconditional generation, but in practical applications researchers are actually interested in more complex conditional tasks, for example motif scaffolding. The extent to which the distilled model can perform these more complex conditional tasks is not described or validated.

**Questions:**

[Q1] Recently there has been a lot of work on all-atomistic structure generation such as La-Proteina, Protpardelle and others. In these cases, scheduling of different low-noise schedules etc can get a lot more complex. Do you think your approaches would generalise to these settings or do you expect further complications there?

[Q2] You show that the one step performance is significantly worse than the multi-step performance due to the missing noise injection. Do you think this is a fundamental limitation of these models or there is a way to get truly one-step generators?

[Q3] As shown in previous works lower temperature sampling restricts you to sampling from a specified subset of your data distribution that has better in silico scores, but with higher temperature you more closely sample from the original data distribution which manifests itself in better FPSD scores etc. Since you distill at a specific temperature, has your model lost the ability to sample from the full data distribution? Is there any way to preserver/recover it?

---

> ### Author Response · Authors · 2025-11-21
> **Response to Reviewer gk6k - Part 1**
>
> We thank Reviewer gk6k for their positive evaluation of our work. We appreciate that the reviewer recognizes the challenges faced when adapting distillation methods to protein backbone generation models with low temperature sampling as well as the practical importance of our sampling acceleration. We would also like to thank the reviewer for considering our iterations of design decisions “a very useful practical resource”. Many of the reviewer’s comments are inspiring to us. In addition to improving the current manuscript, they have sparked ideas for future work that we plan to explore. Below we address the raised questions:
>
> 1. *fold class metrics failure mode*
>
> **Response:** We thank the reviewer for the suggestion. We extracted the outputs of the fold class predictor that Proteina [1] provided, and recorded the counts of each predicted C class. Among the 500 samples generated unconditionally by the pretrained model, the counts and failure rates for each C class are shown in the Table 2 of the main text. Assuming the label mappings are “mainly alpha”, “mainly beta”, “mixed alpha/beta”, “Few secondary structures”, and “Special” from 0 to 4, it seems that the distilled model produces slightly fewer “mainly alpha” and “mainly beta” structures, but slightly more “mixed alpha/beta” structures. The less common cases of “few secondary structures” and “special” also seem to have fewer representations among the distilled-model-generated samples. In terms of failure rate, both the pretrained and distilled model seem to perform worse for “mainly beta” structures, and better for “mainly alpha” and “mixed alpha/beta” structures. For the other two less common classes, there are not enough samples to make any meaningful conclusion. At the A level, the pretrained model produced 15 different A labels while the distilled model produced 13. At the T level, the numbers are 80 and 74. Overall, it does not seem that our fold-class specific metrics being slightly worse are due to underrepresenting a certain fold class.
> In fact, combined with our slightly lower cluster- and TM-based diversity scores, it is expected that our fS scores would be lower since there are fewer fold classes. The lower fS score is probably due to the slightly lower diversity caused by the low temperature sampling. We also want to note that the other two fold class metrics compare the sampling distribution with the PDB and AFDB reference datasets. However, having high distributional similarity to the reference datasets could mean that the samples are not novel. As our distilled model actually produces more novel samples based on the novelty scores, it is expected that the distributional similarity towards the reference datasets decreases.
>
> 2. *motif-scaffolding*
>
> **Response:** We agree with the reviewer that motif scaffolding represents an important and practically valuable direction beyond unconditional generation. Our current work focuses on establishing a strong foundation through data-free distillation, demonstrating that high-quality and efficient protein generation can be achieved without additional supervision. Extending this framework to conditional tasks such as motif scaffolding is an exciting next step, and we plan to explore this direction in future work.
>
> Q1. *Application to all-atom generation models*
>
> **Response:** We thank the reviewer for these insightful questions, which could inspire us and others to explore extending our approach to all-atomistic models such as Protpardelle and La-Proteina. Without experiments, it is difficult to know how readily our method would apply to these settings. For example, we initially expected distilling a protein diffusion model to be straightforward but encountered many failures before arriving at our current simple and effective solution.
>
> **Protpardelle** [2]: Its noise (or step) scale modifies the score during inference in a manner similar to EDM, so we do not anticipate major challenges in handling noise scaling. However, the superposition modeling and dynamically changing masks may introduce additional complications.
>
> **La-Proteina** [3]: This model uses partially latent flow matching with two coupled SDEs and distinct noise schedules, which could require non-trivial adjustments to our SiD-based distillation framework.
>
> We are optimistic that elegant extensions of our current method can be developed for these architectures, though the path may indeed be nontrivial.

---

> ### Author Response · Authors · 2025-11-21
> **Response to Reviewer gk6k - Part 2**
>
> Q2: *Is there a way to get truly one-step generators?*
>
> **Response:** We thank the reviewer for this thoughtful question. Since low-temperature sampling is required by nearly all successful protein generation models, we believe the inability to incorporate noise scaling in a single step remains a clear limitation of one-step generators. Our attempts at directly distilling the teacher into a one-step model were unsuccessful. However, we see promising potential in distilling from our current few-step generators instead, which might serve as a more stable intermediary. This represents an exciting future direction for achieving truly one-step protein generation.
>
> Q3: *Has the model lost the ability to sample from the full data distribution?*
>
> **Response:** We thank the reviewer for this insightful question. As noted in our response to Q2, our generator is trained to match the scores of the teacher network, which approximates the full data distribution, while the noise rescaling is only applied during inference. Setting the noise scale to 1 effectively recovers the full distribution sampling behavior, as also observed in Proteína. Consistent with this, our results (added in Table 1 of our main text) show that sampling with a noise scale of 1 yields lower designability but higher diversity, with improved FPSD, fS, and fJSD metrics indicating closer alignment with the PDB and AFDB distributions.
> We also note that our current distillation framework is data-free, relying solely on self-synthesized samples under the teacher’s guidance. Incorporating real protein data in the future could help further compensate for any residual distribution mismatch, if present. \
> \
> References:
>
> [1] Geffner, T., Didi, K., Zhang, Z., Reidenbach, D., Cao, Z., Yim, J., Geiger, M., Dallago, C., Kucukbenli, E., Vahdat, A., & Kreis, K. (2025). Proteina: Scaling Flow-based Protein Structure Generative Models. The Thirteenth International Conference on Learning Representations. \
> [2] Chu, A. E., Kim, J., Cheng, L., El Nesr, G., Xu, M., Shuai, R. W., & Huang, P.-S. (2024). An all-atom protein generative model. Proceedings of the National Academy of Sciences, 121(27), e2311500121. \
> [3] Geffner, T., Didi, K., Cao, Z., Reidenbach, D., Zhang, Z., Dallago, C., Kucukbenli, E., Kreis, K., & Vahdat, A. (2025). La-Proteina: Atomistic protein generation via partially latent flow matching. arXiv. https://doi.org/10.48550/arXiv.2507.09466

---

### Official Review · Reviewer_FpBX · 2025-11-01

**Soundness:** 3
**Presentation:** 4
**Contribution:** 2
**Rating:** 4
**Confidence:** 4

**Summary:**

This paper presents a significant advancement in the field of protein design, successfully bringing deep generative models closer to practical applications. Through an innovative distillation framework, it effectively addresses the sampling speed bottleneck of existing diffusion and flow-matching models while maintaining high-quality generation. This work holds considerable promise for large-scale protein discovery and engineering applications, opening new avenues for future research in protein design methods. The proposed methodology is clearly articulated, and its effectiveness is thoroughly validated through extensive experiments. Despite minor shortcomings in specific metrics, the overall contribution is positive and substantial.

**Strengths:**

1. The paper successfully adapts score distillation for protein backbone generation, overcoming previous failures. It specifically tackles the challenges of protein structure sensitivity and the need for low-temperature sampling with a novel "few-step distillation + noise scaling" approach, which is a significant methodological contribution.

2.  The most prominent advantage is the dramatic reduction in sampling time—over 20-fold improvement. This is crucial for de novo protein design, which often requires generating thousands to millions of candidate structures for evaluation. This speedup makes large-scale in silico protein design practically feasible and enables tighter integration with iterative design-test exploration cycles.

**Weaknesses:**

1. Limited Scope of Protein Types for Evaluation: The paper primarily focuses on "unconditional protein structure generation and fold class conditional generation" and a single case study for biological plausibility. There is a lack of diverse experimental evaluations across a broader range of protein types, sizes, or specific functional classes. This limited scope makes it challenging to fully ascertain the generalizability and reliability of the proposed distillation framework for real-world protein engineering challenges involving various protein scaffolds or targeted functions.


2. Lack of Detailed Error Analysis for "Undesignable" Structures: The paper notes that one-step generators produce "almost no designable samples" and that even few-step generators need noise scaling to improve designability. While the problem is identified and addressed, a more detailed analysis of the types of structural errors that lead to "undesignable" outcomes in sub-optimal configurations could provide deeper insights into the limitations of distillation for protein backbones and guide future improvements.

3. Dependency on Pre-trained Teacher Model Quality: The entire distillation process relies on the performance of a pre-trained teacher model (Proteína's Mng-tri model). The effectiveness of the distilled generator is inherently capped by the capabilities and potential biases of this teacher. If the teacher model has limitations or exhibits certain failure modes, these could potentially be inherited or even amplified in the distilled, faster generators, which might not be fully explored in the current evaluation.

**Questions:**

None

---

> ### Author Response · Authors · 2025-11-21
> **Response to Reviewer FpBX**
>
> We thank Reviewer FpBX for the thoughtful evaluation. We are encouraged that the review recognizes our work as “a significant advancement in protein design,” with “considerable promise for large-scale applications” and a “clearly articulated, thoroughly validated methodology.” We appreciate these comments and are glad that our approach and findings were viewed positively overall. Given the positive assessment but relatively low rating, we hope our clarifications help bring the overall evaluation into closer alignment with the strengths you highlighted.
>
> 1. *Limited scope of protein types for evaluation*
>
> **Response:** We thank the reviewer for this insightful comment. Our goal was to establish a solid foundation by showing that diffusion distillation can efficiently accelerate high-quality unconditional and fold-conditional protein generation. The distilled model inherits the teacher’s capabilities, which were not fine-tuned for specific functions. We view this as an important first step for protein generation, which is analogous to how LLMs and diffusion vision models evolved from unconditional generation toward instruction-following and goal-conditioned design. We respectfully request that broader functional evaluations be viewed as valuable future work building on this foundation, rather than as a limitation warranting rejection.
>
>
> 2. *Error analysis for “undesignable” structures produced by one-step generators*
>
> **Response:** We have added visualizations of representative “undesignable” structures generated by one-step models in Figure 8, Appendix I. These samples typically lack β-sheets and consist exclusively of α-helices, many of which are unusually long and often oriented in nearly opposite directions. In contrast, our noise-rescaled few-step distillation method largely mitigates these issues, producing more realistic and designable protein backbones.
>
> 3. *Dependency on pretrained teacher model*
>
> **Response:** We thank the reviewer for the thoughtful comment. Our primary goal is to develop a general and efficient distillation framework that accelerates generation without compromising quality, rather than to produce the “best” protein backbone model. The current performance ceiling arises from our data-free distillation setting, where no external data are introduced, and only synthetic samples from the student generator are used for distillation. Thus, the distilled model naturally inherits the teacher’s strengths and limitations.
> Based on the designability, diversity, and novelty metrics, we do not believe the failure modes have been amplified. In fact, our novelty, FPSD, and fJSD metrics suggest that our distilled model produces samples that are less similar to existing structures in the PDB and AFDB compared to the pretrained model. We believe the high designability and novelty is evidence that our distilled model is actually exploring novel modes while successfully avoiding some of the failure modes of the pretrained model.
> The dependency on the teacher is not an inherent limitation of the method; integrating curated protein data or reward-based objectives (e.g., adversarial or post-training alignment losses) could further enhance performance. Such extensions represent promising future directions that build upon the foundation established in this work, where the teacher serves as a knowledgeable source of pretrained priors that provide regularization and guidance, while additional data or reward signals can introduce targeted knowledge to further refine and expand the model’s capabilities.

---

### Author Response · Authors · 2025-11-21
**Overall Response by Authors**

We would like to thank all reviewers for their comprehensive evaluation and feedback. We are encouraged to see that the challenges we faced when adapting the distillation method for protein generation tasks are recognized and the practical contributions of our work are appreciated.

In addressing the reviewers' concerns, we added some experiments and revised the manuscript accordingly. The changes are highlighted in blue for the reviewers' convenience. Below we summarize the changes that we made:
1. We added a clarifying statement in the Introduction that **Proteina [1]  does not have an equivariant architecture**, but still can greatly benefit from distillation.
2. We added **Flow Map Matching** [2] and **Align-Your-Flow** [3] to the Related Work section.
3. We provided additional details about the **fake score loss**, **$\alpha$**, and **intermediate loss terms** before arriving at the final generator loss in Section 3.1.
4. We clarified that during training, when generating $\boldsymbol{x}_g$, **all except the selected step $k$** has the stop gradient operation.
5. We **sampled and evaluated from the full distribution** using our 16-step distilled model by setting the noise scale to 1 and reported the results as an additional row in Table 1.
6. We **analyzed the failure modes** of our unconditional generation results by classifying them into fold classes and comparing fold-class-specific failure rates and counts. The results in Table 2 have cleared our doubts about underrepresenting certain fold classes. We now believe that the drop in fold-class-related metrics are likely to be byproducts of our lower diversity among generated batches and higher novelty compared to reference datasets.
7. We **applied our distillation method to Genie** [4], a diffusion model based on an equivariant network. The results are shown in Table 4, Section 4.7 and prove that our distillation framework works with both diffusion- and flow-based models, either with or without the equivariant network constraint.
8. We added visualizations and **analysis on our generated structures that were undesignable** in Figure 8, Appendix I.

We believe these additions and modifications further improves our work and we thank the reviewers for their suggestions.

References:
[1] Geffner, T., Didi, K., Zhang, Z., Reidenbach, D., Cao, Z., Yim, J., Geiger, M., Dallago, C., Kucukbenli, E., Vahdat, A., & Kreis, K. (2025). Proteina: Scaling Flow-based Protein Structure Generative Models. The Thirteenth International Conference on Learning Representations.
[2] Boffi, N. M., Albergo, M. S., & Vanden-Eijnden, E. (2025). Flow map matching with stochastic interpolants: A mathematical framework for consistency models. arxiv preprint arxiv:2406.07507
[3] Sabour, A., Fidler, S., & Kreis, K. (2025). Align Your Flow: Scaling Continuous-Time Flow Map Distillation. arXiv preprint arXiv:2506.14603
[4] Lin, Y., & AlQuraishi, M. (2023). Generating Novel, Designable, and Diverse Protein Structures by Equivariantly Diffusing Oriented Residue Clouds. In Proceedings of the 40th International Conference on Machine Learning (pp. 20978-21002). PMLR.

---

### Author Response · Authors · 2025-12-03
**Rebuttal Summary by Authors**

In lieu of recent events, we would like to summarize our rebuttal efforts for the Area Chair. Following the initial reviewer comments, which gave scores of 8, 6, 4, and 4, we worked diligently to address the concerns by running additional experiments, clarifying methodological details in the manuscript, and answering reviewer questions directly in our responses. We are encouraged that Reviewer bFFC found our clarifications helpful and subsequently increased their score from 4 to 6, leading to an overall score of 8, 6, 6, and 4. Although we did not have the opportunity to engage further with the remaining reviewers, we ensured that all comments were addressed carefully and thoroughly. A detailed list of the revisions made to the manuscript can be found in our overall response. We hope this summary can help the Area Chair make an objective and unbiased assessment.

---

### Meta-Review · Area_Chair_aPEU · 2026-01-06

**Summary:**

In this submission, the authors leverage score distillation to accelerate diffusion/flow-based protein backbone generation models, thereby verifying the potential of this technique to accelerate protein design while maintaining designability and diversity scores.

The main concerns of the reviewers are about 1) the solidity of the experimental part, and 2) the writing of this paper. In the rebuttal phase, the authors provided more explanations. However, without sufficient analytic experiments, the concerns are not fully resolved.

1. In my opinion, the aim of this work is to accelerate protein design (rather than introducing new/novel techniques), which is totally fine. However, given this aim, it is necessary to compare the proposed method with other strong acceleration strategies, e.g., the rectified flow used in [A]. Without strong baselines, the advantage of score distillation is not convincing. Notably, flow matching and diffusion can be converted into one another (as shown in EDM [B]), so there is no difficulty (or unfairness) in comparing the proposed method with flow rectification.

2. Existing methods like FrameFlow and FrameDiff provide detailed analysis of their performance in generating proteins of different lengths. Following these methods, the authors should analyze the impact of distillation in generating proteins of different lengths.

3. Moreover, for the acceleration methods (including distillation and flow rectification), evaluating their impacts on data distribution drifting is necessary.

Considering the above issues, I think this work requires one more round of review.

[A] Yue, A., Wang, Z., & Xu, H. ReQFlow: Rectified Quaternion Flow for Efficient and High-Quality Protein Backbone Generation. ICML, 2025.

[B] Tero Karras, Miika Aittala, Samuli Laine, and Timo Aila. Elucidating the design space of diffusion-based generative models. NeurIPS, 2022.

**Reviewer Concerns:**

I think the solidity of the experimental part of this work is questionable, and this concern is still outstanding after I read the paper and the discussions.

**Reviewer Scores:**

Although Reviewer bFFC increased his/her score in the rebuttal phase, I think Reviewer gk6k could have reduced his/her score, and the remaining reviewers would have maintained their original scores if they had sufficiently discussed with each other.

---

### Decision · Program_Chairs · 2026-01-26

Reject